# Ordered clustering of single atomic Te vacancies in atomically thin PtTe$_2$ promotes hydrogen evolution catalysis

Xinzhe Li [1,2,8], Yiyun Fang[1,2,3,8], Jun Wang[1,4,8], Hanyan Fang [2,8], Shibo Xi[5], Xiaoxu Zhao[6], Danyun Xu[1], Haomin Xu [2], Wei Yu [2], Xiao Hai [2], Cheng Chen[2], Chuanhao Yao [2,3], Hua Bing Tao[7], Alexander G. R. Howe[6], Stephen J. Pennycook [6], Bin Liu [7✉], Jiong Lu [2✉] & Chenliang Su [1✉]

Exposing and stabilizing undercoordinated platinum (Pt) sites and therefore optimizing their adsorption to reactive intermediates offers a desirable strategy to develop highly efficient Pt-based electrocatalysts. However, preparation of atomically controllable Pt-based model catalysts to understand the correlation between electronic structure, adsorption energy, and catalytic properties of atomic Pt sites is still challenging. Herein we report the atomically thin two-dimensional PtTe$_2$ nanosheets with well-dispersed single atomic Te vacancies (Te-SAVs) and atomically well-defined undercoordinated Pt sites as a model electrocatalyst. A controlled thermal treatment drives the migration of the Te-SAVs to form thermodynamically stabilized, ordered Te-SAV clusters, which decreases both the density of states of under-coordinated Pt sites around the Fermi level and the interacting orbital volume of Pt sites. As a result, the binding strength of atomically defined Pt active sites to H intermediates is effectively reduced, which renders PtTe$_2$ nanosheets highly active and stable in hydrogen evolution reaction.

[1] SZU-NUS Collaborative Center and International Collaborative Laboratory of 2D Materials for Optoelectronic Science and Technology of Ministry of Education, Institute of Microscale Optoelectronics, Shenzhen University, Shenzhen, Guangdong, China. [2] Department of Chemistry, National University of Singapore, Singapore, Singapore. [3] Frontiers Science Center for Flexible Electronics, Xi'an Institute of Flexible Electronics (IFE) and Xi'an Institute of Biomedical Materials & Engineering, Northwestern Polytechnical University, Xi'an, China. [4] School of Electrical Engineering and Automation, Wuhan University, Wuhan, China. [5] Institute of Chemical and Engineering Sciences, Agency for Science, Technology and Research (A*STAR), Singapore, Singapore. [6] Department of Materials Science and Engineering, National University of Singapore, Singapore, Singapore. [7] School of Chemical and Biomedical Engineering, Nanyang Technological University, Singapore, Singapore. [8] These authors contributed equally: Xinzhe Li, Yiyun Fang, Jun Wang, Hanyan Fang. ✉email: liubin@ntu.edu.sg; chmluj@nus.edu.sg; chmsuc@szu.edu.cn

Global challenges, such as the increased energy demand, environmental pollution, and limited earth resources, are currently threatening sustained human development. Exploiting advanced materials and green technology, e.g., electrocatalysis, to convert sustainable resources (e.g., $H_2O$, $CO_2$, $N_2$, and solar energy) into high value-added products (e.g., $H_2$, $O_2$, hydrocarbons, and $NH_3$) is promising to address these problems[1]. For example, transition metal phosphides exhibit platinum (Pt)-like activity for hydrogen production via electrocatalytic water splitting[2–4]. However, up to now, rational design and preparation of advanced electrocatalysts with high activity and stability is still urgently required for practical applications, especially for Pt-based catalysts, which play versatile roles in energy-related electrocatalysis, such as alcohol oxidation reaction, hydrogen evolution/oxidation reaction (HER/HOR), and oxygen reduction reaction (ORR)[5]. To obtain excellent performance in these reactions, the catalytic behavior of atomic Pt sites, along with their local-structure environments needs to be deciphered. In this regard, over the past decades, a large variety of strategies have been developed to identify the decisive factors that influence the catalytic behaviors of Pt sites during various electrochemical reactions[6–16]. The corresponding results unveil that exposing undercoordinated Pt sites as well as optimizing the adsorption of the reaction intermediates (e.g., $H^*$, $O^*$, $OH^*$, $OOH^*$) to these sites is a desirable strategy to drastically enhance electrocatalytic activity[6–13]. For instance, acid-etched Pt-Ni alloys provide sufficient and accessible Pt sites, and the electronic and strain effects help to weaken the binding strength of Pt sites to oxygenated species, thus showing excellent ORR activities[6]. Although great progress has been achieved, simultaneously realizing the exposure and stabilization of undercoordinated Pt sites, as well as optimizing the relationship between electronic structure, adsorption energy, and catalytic properties of Pt sites at the atomic scale still remains a grand challenge, in part due to difficulties in precisely tailoring such kind of well-defined atomic Pt sites.

In this work, we design and prepare atomically thin two-dimensional (2D) $PtTe_2$ nanosheets (NSs) with well-dispersed single atomic Te vacancies (Te-SAVs) by electrochemically exfoliating bulk $PtTe_2$ crystals, in which large numbers of atomically defined undercoordinated and stabilized Pt sites are exposed. Consequently, the obtained 2D $PtTe_2$ NSs have a well-defined structure and can serve as a Pt-based model catalyst, which are different from traditional Te or Pt-based noble metal materials. The following heat treatment causes migration of the random Te-SAVs to form ordered Te-SAV clusters. Both electrochemical measurements and density functional theory (DFT) calculations show that the heat treatment-induced migration of Te-SAVs in $PtTe_2$ can effectively tailor the hydrogen adsorption energy ($\Delta G_{H^*}$) on the undercoordinated Pt sites. Consequently, the $PtTe_2$ NSs with ordered clusters of Te-SAVs exhibit much-enhanced HER activity with an exceptionally low onset potential (~0 mV), overpotential ($\eta$, 22 mV at 10 mA cm$^{-2}$) and Tafel slope (29.9 mV per dec$^{-1}$). Furthermore, after 20,000 continuous potential cycles and chronopotentiometry test at high current densities (200 mA cm$^{-2}$) for 24 h, the catalyst displays negligible activity decay, outperforming the benchmark Pt/C catalyst.

## Results

**Preparation of atomically thin $PtTe_2$ NSs.** $PtTe_2$ crystallizes in a $CdI_2$-type trigonal (1 T) structure (P$\bar{3}$m1, a = b = 4.026 Å, c = 5.221 Å) with adjacent layers (interlamellar space: 3.52 Å) connected via weak Van der Waals interaction (Supplementary Fig. 1)[17]. In this work, a chemical vapor transport (CVT) technique was employed to synthesize bulk $PtTe_2$ crystals with closely stacked lamellar architecture (Supplementary Fig. 2), following by

electrochemical exfoliation (detailed in Supplementary Figs. 3 and 4) to produce atomically thin 2D $PtTe_2$ NSs as illustrated in Fig. 1a. Notably, a wealth of Te-SAVs was generated during the crystal growth period (discussed in the following part). The exfoliated $PtTe_2$ NSs display broad (101), (102), (110), and (201) diffraction peaks in the X-ray diffraction (XRD) pattern (Fig. 1c). The disappearance of strong and sharp diffraction peaks results from loss of long-range order in $PtTe_2$ after exfoliation[18,19]. Atomic force microscopy (AFM) measurements show that the $PtTe_2$ NSs have lateral dimensions of ~1–15 µm with thicknesses of ~0.6–6 nm (average thickness: ~3 nm) (Fig. 1d and Supplementary Fig. 5). To examine the morphology, atomic structure, and chemical composition of the exfoliated $PtTe_2$ NSs, microscopy characterizations including transmission electron microscopy (TEM), and aberration-corrected high-angle annular dark-field scanning TEM (HAADF-STEM) were performed. Figure 1e displays a low magnification HAADF-STEM image, showing the corrugated and roughened 2D surface of few-layer $PtTe_2$ NSs. Notably, the high-resolution HAADF-STEM image (Fig. 1f), viewed from the [001] zone axis, displays atomic lattice composing of alternating bright and dark spots, which correspond, respectively, to the Pt and Te atomic columns as indicated in the atomic model (inset at the lower right corner of Fig. 1f). Furthermore, vacancies can be directly visualized with distinguishable contrast in the few-layer $PtTe_2$ NSs (inset at the lower left corner of Fig. 1f). The corresponding line intensity profile (inset in the upper portion of Fig. 1f) combined with the atomic model (inset at the lower right corner of Fig. 1f) indicates that the point defect corresponds to Te vacancy. Additionally, the elemental mapping shows a homogenous distribution of Pt and Te elements over the whole $PtTe_2$ NSs (Fig. 1g). No carbon signal is detected on the surface of exfoliated $PtTe_2$ NSs, indicating intercalator and/or its decomposition products can be washed away after exfoliation. Such a conclusion can be further supported by the elemental analysis result, which reveals that the content of carbon in the exfoliated $PtTe_2$ NSs is about zero.

**Thermal induced migration of Te-SAVs in $PtTe_2$ NSs.** Prior to the heat treatment, the thermal stability of $PtTe_2$ NSs was investigated by thermogravimetry analysis (TGA) at ambient pressure, and the results (Supplementary Fig. 6) indicate high thermal stability of $PtTe_2$ NSs up to ~650 °C. Accordingly, $PtTe_2$ NSs were heated to 200, 400, and 600 °C, respectively, in Ar (gas flowrate: 200 sccm) for 1 h, yielding $PtTe_2$-200 NSs, $PtTe_2$-400 NSs, and $PtTe_2$-600 NSs. Intriguingly, based on the result of inductively coupled plasma-optical emission spectroscopy (ICP–OES; Perkin Elmer Avio 500, UK), the $PtTe_2$ NSs keep a constant atomic ratio of Pt/Te during the entire heat treatment process (Supplementary Table 1, Supplementary Fig. 7). Additionally, other characterizations including XRD (Supplementary Fig. 8), TEM (Supplementary Figs. 9–11), and X-ray photoelectron spectroscopy (XPS, Supplementary Fig. 12) all show negligible changes in morphology and composition of $PtTe_2$ NSs during heat treatment.

To show heat treatment-induced migration of Te-SAVs in $PtTe_2$ NSs, scanning tunneling microscopy (STM) was performed to better visualize the evolution of Te-SAVs as a function of annealing temperature. Prior to heat treatment, as shown in Fig. 2a, a high density of uniformly dispersed SAVs (black dots) are observed on the surface of thin $PtTe_2$ sheets. Figure 2b displays the atomically resolved STM image of three individual Te-SAVs on the surface. A superimposition of the atomic structure over the close-up STM image reveals that atomic vacancies reside at the Te sites. This conclusion can be further supported by ICP–OES and HAADF-STEM results mentioned

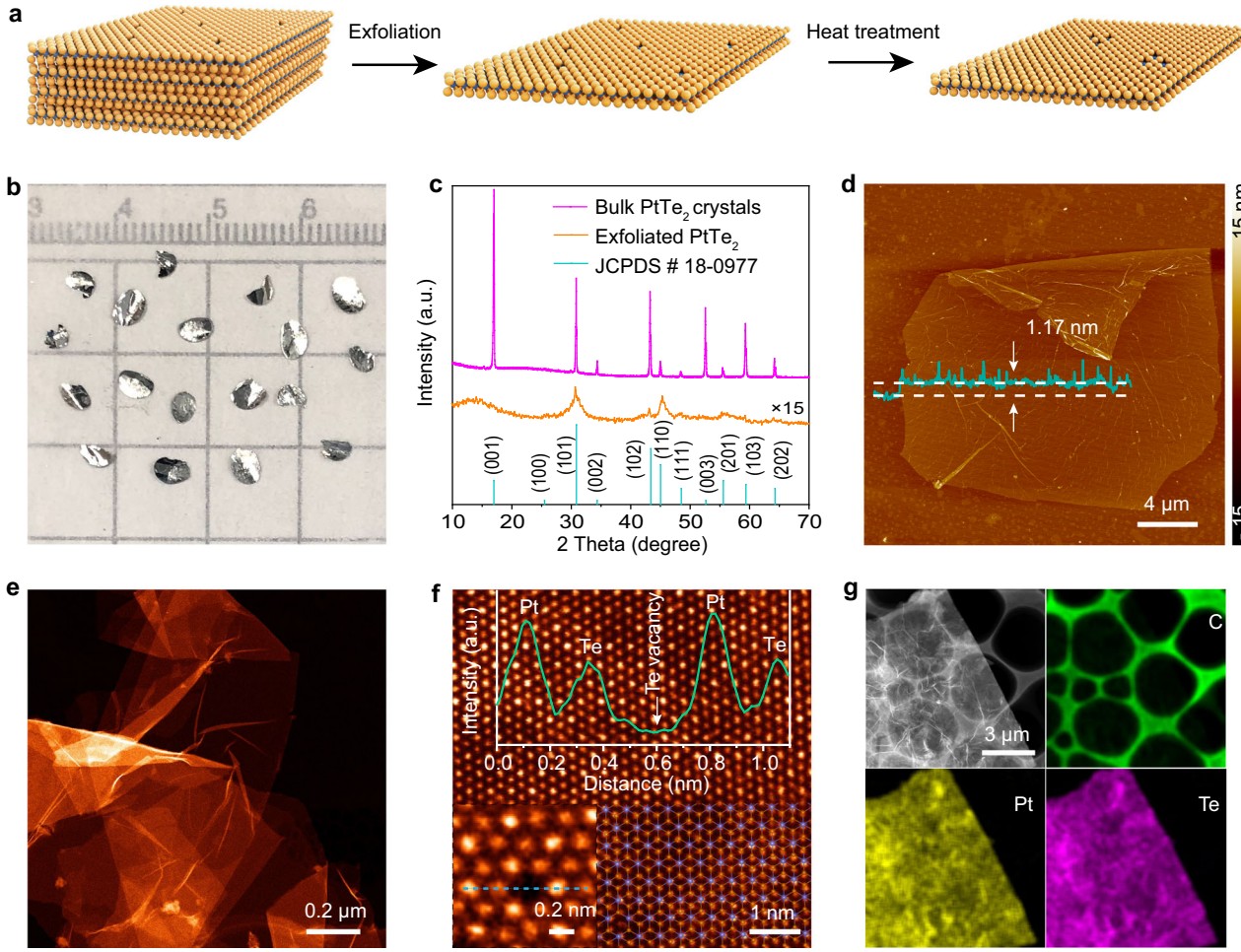

**Fig. 1 Preparation and characterization of PtTe₂ NSs. a** Schematic illustration showing the fabrication of atomically thin PtTe₂ NSs with different vacancy structures. **b** Digital photograph of the PtTe₂ crystals on a millimeter-grade paper. **c** XRD patterns of bulk PtTe₂ crystals and exfoliated PtTe₂ NSs. The standard XRD pattern of PtTe₂ (JCPDS # 18-0977) is shown as a reference. **d** AFM image of atomically thin PtTe₂ NSs. **e** HAADF-STEM image of few-layer PtTe₂ NSs. **f** High-resolution HAADF-STEM image of few-layer PtTe₂ NSs and the corresponding structural model of PtTe₂. Inset on top shows the intensity profile corresponding to the blue line. **g** HAADF-STEM image and the corresponding elemental mapping images of exfoliated PtTe₂ NSs.

above as well as other reported transition metal dichalcogenide systems[20,21], demonstrating the vacancies are single atomic Te vacancies. In addition, statistical analysis of STM results in Fig. 2c shows that all Te-SAVs are separated by a certain distance from about 0.8 to about 2.4 nm. Fascinatingly, the heat treatment process ubiquitously induces migration of Te-SAVs in PtTe₂ to form ordered trigonal Te-SAV clusters on the surface, as highlighted by red triangle dotted box in Fig. 2d. In addition, these trigonal Te-SAV clusters become dominant after heat treatment. By superimposing the atomic model on the close-up STM image in Fig. 2e, it can be seen that the cluster consists of three Te-SAVs, each of which is still separated by a Te atom. Figure 2f analyzes the distance between adjacent Te-SAVs after heat treatment. As expected, the distance of two neighboring Te-SAVs decreases significantly after forming clusters. Thus, different from traditional works that mainly focus on regulating the number of vacancies and doping vacancies via heteroatoms[22], the obtained PtTe₂ exhibits a well-defined atomic structure, which can be served as a model catalyst to investigate the influence of vacancy evolution on the catalytic performance.

The chemical states of PtTe₂ were examined by XPS. As shown in Supplementary Fig. 12, for bulk PtTe₂ crystals, the core-level Pt 4$f$ XPS spectrum exhibits two peaks at binding energies of 71.1 (4$f_{7/2}$) and 74.4 (4$f_{5/2}$) eV, which can be assigned to metallic

Pt[23]. Two zero valent state peaks associated to Te⁰ at 572.0 (3$d_{5/2}$) and 582.3 eV (3$d_{3/2}$), together with two oxidation state peaks associated to Te$^{IV}$ arising from the Te = O bonds after exposing PtTe₂ crystals in air, are detected in Te 3$d$ XPS spectrum[23]. After exfoliation and heat treatment, both Pt and Te in PtTe₂ NSs, PtTe₂-200 NSs, PtTe₂-400 NSs, and PtTe₂-600 NSs still mainly exist in their metallic states, and the gradual appearance of positive binding energy shift of Pt 4$f_{7/2}$, Pt 4$f_{5/2}$, Te 3$d_{5/2}$, and Te 3$d_{3/2}$ with increasing heat treatment temperature may result from the influence of improved oxygen adsorption after air exposure or electron transfer between Pt and Te atoms. To avoid exposing the PtTe₂ NSs to air and therefore provide a definitive study of the electronic structure and coordination environment of Pt in exfoliated PtTe₂ NSs during thermal treatment, in situ X-ray absorption near-edge structure (XANES) spectra of Pt L₃ edge were collected in a home-made testing setup from room temperature to 600 °C in He gas[24]. The results (Fig. 2g) show that with increasing heat treatment temperature, a slight positive shift of Pt L₃ edge of XANES is observed, indicating a slight increase in average valance state of Pt in PtTe₂ NSs during Te-SAVs migration. After cooling down to room temperature in He, the Pt L₃ edge XANES spectrum of PtTe₂-600 NSs was collected and compared with pristine PtTe₂ NSs, and Pt foil. As revealed in Fig. 2h, the edge energy of pristine PtTe₂ NSs is the

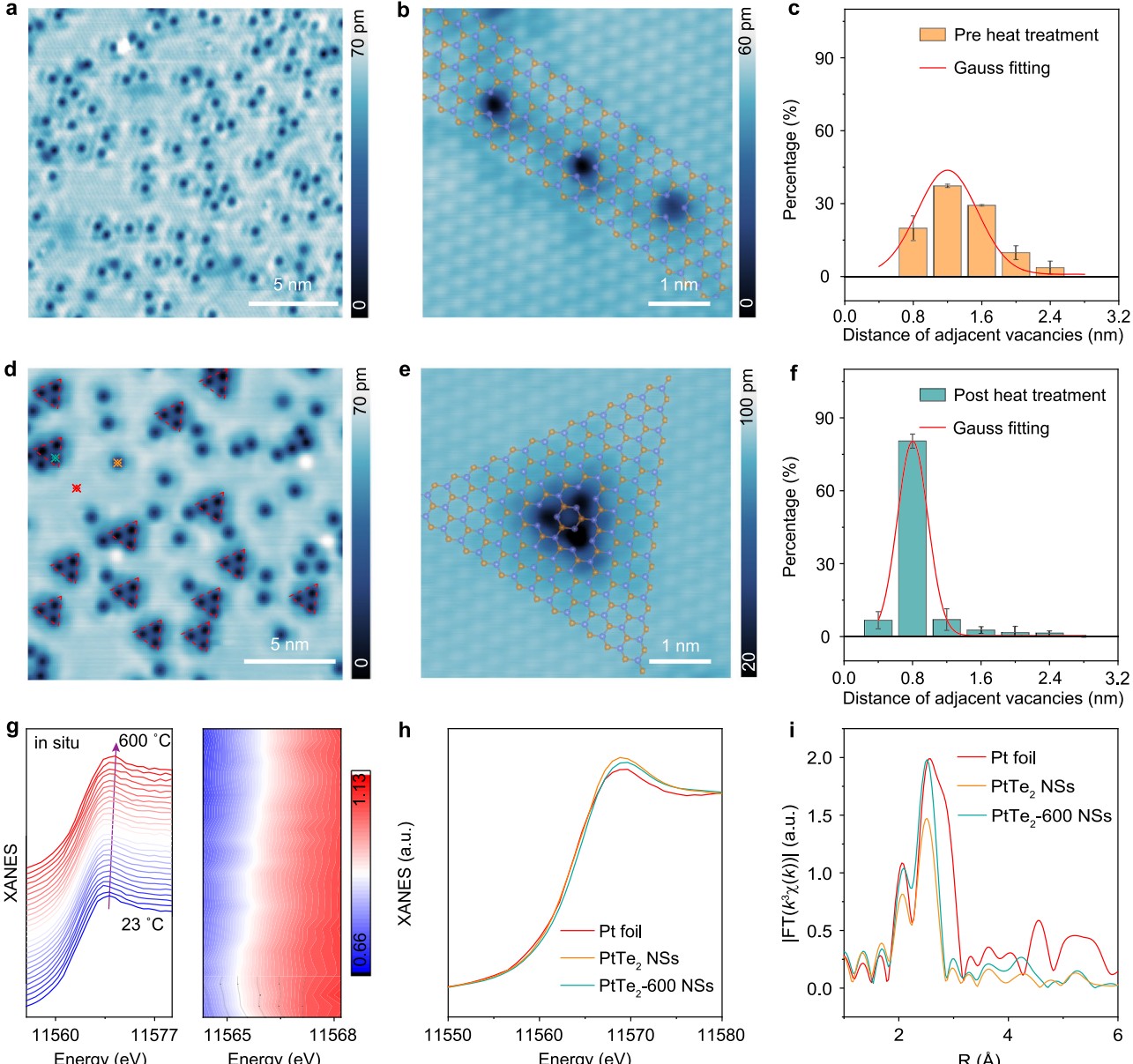

**Fig. 2 Heat treatment-induced migration of Te-SAVs in PtTe$_2$. a** Large-scale STM image of thin PtTe$_2$ sheets taken at $V_S = -2.0$ V and $I = 30$ pA. **b** Close-up STM image showing three individual Te-SAVs on the surface of thin PtTe$_2$ sheets with the corresponding structural model superimposed. The yellow and blue atoms represent Te and Pt atoms, respectively. Te atoms at the bottom of PtTe$_2$ structural model are omitted. **c** Statistic distance of neighboring Te-SAVs in thin PtTe$_2$ sheets before heat treatment. **d** Large-scale STM image of thin PtTe$_2$ sheets after heat treatment. **e** Close-up STM image showing the formation of ordered trigonal Te-SAV clusters on the surface of thin PtTe$_2$ sheets after heat treatment with the corresponding structural model superimposed. Te atoms at the bottom of PtTe$_2$ structural model are omitted. **f** Statistic distance of neighboring Te-SAVs in thin PtTe$_2$ sheets after heat treatment. The error bars in **c** and **f** represent standard deviation of different technical replicates. **g** Stacking plots of the in situ Pt L$_3$-edge XANES spectra of PtTe$_2$ NSs collected from room temperature to 600 °C in He gas (left), and the corresponding 2D contour plots of the in situ Pt L$_3$-edge XANES spectra as shown in **g** (right), revealing the positive shift of the white line. **h** Normalized Pt L$_3$-edge XANES spectra for PtTe$_2$ NSs, PtTe$_2$-600 NSs, and Pt foil. **i** Fourier-transformed $k^3$-weighted EXAFS spectra for PtTe$_2$ NSs, PtTe$_2$-600 NSs, and Pt foil.

same as that of Pt foil, suggesting that the Pt atoms in PtTe$_2$ NSs are mainly present in zero valance state, which is in accordance with the XPS results. However, the Pt L$_3$ absorption edge of PtTe$_2$-600 NSs shows a slight positive shift compared with that of the pristine PtTe$_2$ NSs, revealing that clustering of Te-SAVs in PtTe$_2$ NSs would lead to a decrease in the number of Pt electrons. Additionally, L$_3$-edge XANES spectra of Te in PtTe$_2$ samples were collected and compared in Supplementary Fig. 13a. As expected, the Te absorption edge position of PtTe$_2$-600 NSs is lower in energy than that of PtTe$_2$ NSs, indicating decreased average

valance state of Te in PtTe$_2$-600 NSs. The above results indicate electron transfer from Pt to Te atoms in PtTe$_2$ NSs during thermal treatment. In addition, both PtTe$_2$ NSs and PtTe$_2$-600 NSs show higher white line intensities of the Pt L$_3$-edge as compared to Pt foil, suggesting a higher density of state of $d$ band vacancy for Pt atoms in PtTe$_2$ than that for metallic Pt[25]. Figure 2i shows the corresponding Fourier-transformed (FT) $k^3$-weighted extended X-ray absorption fine structure (EXAFS) spectrum for PtTe$_2$ NSs, PtTe$_2$-600 NSs, and Pt foil. It can be seen that the EXAFS spectrum for PtTe$_2$ NSs and PtTe$_2$-600 NSs

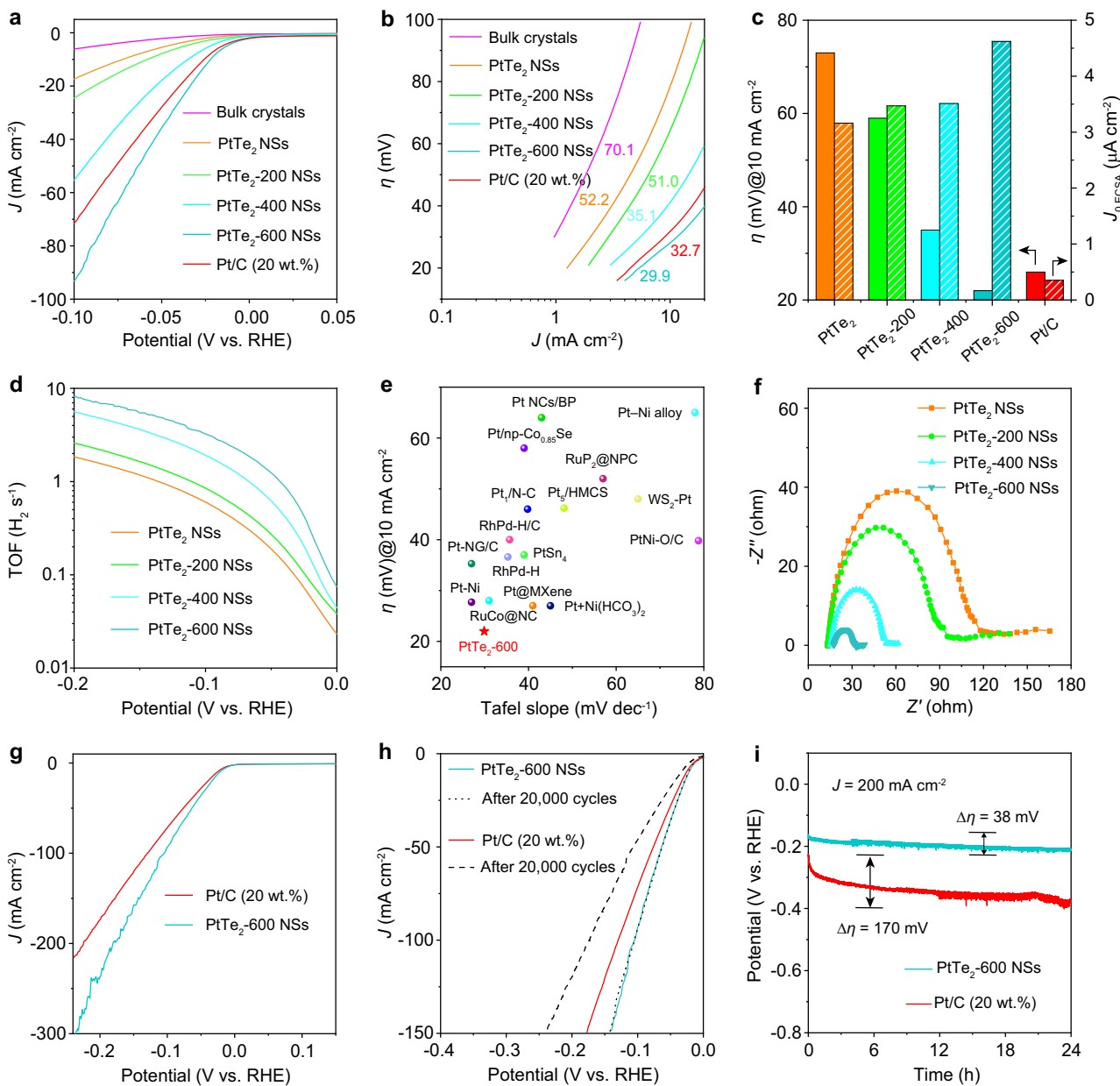

**Fig. 3 HER performance. a** LSV curves of bulk PtTe$_2$ crystals, PtTe$_2$ NSs, PtTe$_2$-200 NSs, PtTe$_2$-400 NSs, PtTe$_2$-600 NSs, and Pt/C recorded in 1.0 M KOH at a scan rate of 5 mV s$^{-1}$, with 80 % iR compensation. **b** The corresponding Tafel plots. **c** Comparison of overpotential at 10 mA cm$^{-2}$ and $J_{0\,ECSA}$. **d** The potential-dependent TOF curves for PtTe$_2$ NSs, PtTe$_2$-200 NSs, PtTe$_2$-400 NSs, and PtTe$_2$-600 NSs. **e** Performance comparison of PtTe$_2$-600 NSs with other reported noble metal-based HER catalysts with high activity in 1.0 M KOH. The detailed information of these catalysts was listed in Supplementary Table 2. **f** Nyquist plots collected at an overpotential of 100 mV. **g** Comparison of LSV curves for PtTe$_2$-600 NSs and Pt/C at large current density. **h** LSV curves of PtTe$_2$-600 NSs and Pt/C before and after 20,000 cycles of accelerated CV test. **i** Chronopotentiometry measurement recorded at 200 mA cm$^{-2}$ for 24 h without iR compensation.

display similar coordination environment except that the amplitude of the first shell peak (at 2.51 Å) for PtTe$_2$-600 NSs is obviously enhanced, which could be due to the decreased local-structure disorder around Pt atom after migration of Te-SAVs to form Te-SAV clusters. Such a phenomenon can be further displayed by the enhanced amplitude from PtTe$_2$ NSs to PtTe$_2$-600 NSs in the EXAFS curves in $K$ space (Supplementary Fig. 13b), and the results of EXAFS fitting, where the disorder degree of PtTe$_2$-600 NSs is 0.0028 ± 0.0004, much lower than that of PtTe$_2$ NSs (0.0037 ± 0.0007).

**Properties of PtTe$_2$ in HER.** The electrocatalytic performances of various PtTe$_2$ samples were evaluated in both acidic and alkaline

electrolyte. As displayed in the linear sweep voltammetry (LSV) curves in Fig. 3a, PtTe$_2$ NSs show substantially larger geometric HER current density ($J$) than bulk PtTe$_2$ crystals at the same overpotential resulting from the more exposed active sites of PtTe$_2$ NSs after exfoliation[19,26]. Interestingly, the HER activity is gradually enhanced from PtTe$_2$ NSs to PtTe$_2$-600 NSs because of the accelerated catalytic kinetics as shown in the corresponding Tafel slopes (Fig. 3b, Supplementary Fig. 14)[27]. The PtTe$_2$-600 NSs feature an exceptionally low HER onset potential (~0 mV, Supplementary Fig. 15) and overpotential (22 mV at $\eta$ = 10 mA cm$^{-2}$, left of Fig. 3c), superior to the state-of-the-art 20 wt.% Pt/C catalyst (26 mV at $\eta$ = 10 mA cm$^{-2}$, left of Fig. 3c). To further display the much-enhanced catalytic

activity, the exchange current density ($J_0$) based on electrochemically active surface area (ECSA) was calculated by extrapolating the Tafel plot (Supplementary Fig. 14)[28–30]. It shows that $J_{0\ ECSA}$ increases in the order of PtTe$_2$ NSs (3.16 μA cm$^{-2}$) < PtTe$_2$-200 NSs (3.46 μA cm$^{-2}$) < PtTe$_2$-400 NSs (3.51 μA cm$^{-2}$) < PtTe$_2$-600 NSs (4.62 μA cm$^{-2}$), while $J_{0\ ECSA}$ of Pt/C is 0.35 μA cm$^{-2}$ (right of Fig. 3c). Additionally, the TOF trend (measured at −0.2 V vs. RHE) also follows the same order with PtTe$_2$ NSs (1.84 s$^{-1}$) < PtTe$_2$-200 NSs (2.59 s$^{-1}$) < PtTe$_2$-400 NSs (5.59 s$^{-1}$) < PtTe$_2$-600 NSs (8.21 s$^{-1}$) (Fig. 3d). Considering that PtTe$_2$-600 NSs show similar composition with PtTe$_2$ NSs (ICP–OES results), PtTe$_2$-600 NSs exhibit much-enhanced intrinsic HER catalytic activity, outperforming most of the reported highly active HER catalysts (Fig. 3e, and Supplementary Table 2). The high HER activity of the PtTe$_2$-600 NSs can be further supported by the electrochemical impedance spectroscopy (EIS) results (Fig. 3f, Supplementary Fig. 16), which display much reduced charge transfer resistance compared to the PtTe$_2$ NSs[31]. Even at large current densities, the PtTe$_2$-600 NSs still display superior HER activity than commercial Pt/C (Fig. 3g). The LSV curves based on mass activity of Pt in PtTe$_2$-600 NSs and Pt/C are shown in Supplementary Fig. 17. The mass activity of Pt in PtTe$_2$-600 NSs is obviously higher than that in Pt/C at the same potential. At −0.2 V vs. RHE, the mass activity of Pt in PtTe$_2$-600 NSs is calculated to be 1.55 A mg$_{Pt}^{-1}$, while that is only 1.13 A mg$_{Pt}^{-1}$ for Pt in Pt/C. Apart from the activity, catalytic stability is equally important in practical applications. Accelerated cyclic voltammetry (CV) test elucidates the robustness of PtTe$_2$-600 NSs in HER catalysis with virtually unchanged polarization curves even after 20,000 CV cycles (Fig. 3h). Furthermore, long-term chronopotentiometry measurement shows a slight change of overpotential after 24 h at a constant current density of 200 mA cm$^{-2}$ for PtTe$_2$-600 NSs, much better than that of Pt/C catalyst (Fig. 3i). The PtTe$_2$-600 NSs after long-term stability test were characterized, showing negligible change of morphology, structure, and composition (Supplementary Fig. 18). The similar catalytic trend of PtTe$_2$ for HER was also demonstrated in the acidic electrolyte, as shown in Supplementary Fig. 19.

**Theoretical insights between vacancy clustering and HER.** To shed light on the influence of ordered clustering of Te-SAVs in atomically thin PtTe$_2$ on HER activity, a series of theoretical calculations were performed based on the first principle methods. When the thickness increases from monolayer to bilayer, PtTe$_2$ evolves from semiconductive (bandgap: ~0.84 eV) to metallic. The electronic structure of trilayer PtTe$_2$ has already resembled to its bulk material (Supplementary Figs. 20 and 21). Since metallic nature is of pivotal importance for hydrogen adsorption, the bilayer PtTe$_2$ is expected to show good HER activity, which can be confirmed by calculating the hydrogen adsorption energy at active sites in different layered PtTe$_2$ (Supplementary Fig. 22). To reduce the computational cost, the metallic bilayer PtTe$_2$ was adopted as the theoretical model for the following calculations.

To uncover the dependence of clustering degree of Te vacancies on the heat treatment temperature, the relative stability of bilayer PtTe$_2$ with different types of Te vacancies was investigated by calculating their formation energy ($E_f$)[32]. As displayed in Supplementary Fig. 23, the bilayer PtTe$_2$ models, containing three Te vacancies and separated by different numbers of Te atoms, were constructed to model the different structures of PtTe$_2$ NSs. For simplicity, herein, we use PtTe$_2$-xTe (x = 3, 2, 1, or 0), where "x" refers to the number of Te atoms between two isolated Te vacancies on the surface of bilayer PtTe$_2$, to denote the structure. Specifically, the PtTe$_2$-3Te represents the exfoliated PtTe$_2$ NSs with isolated Te-SAVs, the PtTe$_2$-1Te represents the

PtTe$_2$-600 NSs with plenty of trigonal Te-SAVs, while the PtTe$_2$-2Te represents the intermediate transition state of PtTe$_2$ NSs during heat treatment, where the distance between two Te-SAVs is closer than that of Te-SAVs in PtTe$_2$ NSs. Figure 4a shows the calculated $E_f$ based on the energy difference between the intact PtTe$_2$ and defective PtTe$_2$-xTe. The $E_f$ decreases in the order of PtTe$_2$-1Te < PtTe$_2$-2Te < PtTe$_2$-3Te ≪ PtTe$_2$-0Te, suggesting that heat treatment would lead to formation of PtTe$_2$-1Te because of the lowest $E_f$, which explains the formation of ordered trigonal Te-SAV clusters in PtTe$_2$-600 NSs (Fig. 2e). In addition, when Te-SAVs coalesce into a larger vacancy, the $E_f$ greatly increases, indicating the structure of PtTe$_2$-0Te hardly exists in the sample. This unstable structure may be ascribed to the formation of lower coordinated Pt atoms.

The step of H$_2$O dissociation is considered in alkaline solution, which may exert extra barrier on the HER process[33]. As shown in Supplementary Fig. 24, the barrier of H$_2$O dissociation on defective PtTe$_2$ (1.09 eV) is very close to that on Pt (111) surface (1.07 eV), indicating that H$_2$O dissociation has a negligible impact on the activity difference between PtTe$_2$ and Pt for HER. The HER performance of PtTe$_2$-xTe was then evaluated by $\Delta G_{H^*}$, where a higher $\Delta G_{H^*}$ indicates a weaker hydrogen adsorption, and vice versa. An ideal HER catalyst should bind to hydrogen neither too strongly nor too weakly, giving rise to $\Delta G_{H^*}$ close to zero[34]. The calculated hydrogen adsorption energies reveal that the undercoordinated Pt sites rather than Te sites in PtTe$_2$-1Te are the active sites (Supplementary Fig. 25), which was further verified by the poison experiments (Supplementary Fig. 26). Subsequently, the location of different undercoordinated Pt sites was investigated, denoted as PtTe$_2$-Pty (y = 1, 2, or 3), where "y" is the specific atomic position of Pt atoms (Supplementary Fig. 23). Figure 4b shows the calculated $\Delta G_{H^*}$ on different undercoordinated Pt sites in PtTe$_2$-xTe and the Pt site with the lowest |$\Delta G_{H^*}$| was chosen as the dominate active site of the corresponding catalysts. It can be observed that the $\Delta G_{H^*}$ for PtTe$_2$-xTe varies in the following order: PtTe$_2$-3Te (−0.044 eV) < PtTe$_2$-2Te (-0.027 eV) < PtTe$_2$-1Te (−0.013 eV) > PtTe$_2$-0Te (−0.028 eV). In addition, the Pt1 and Pt3 sites in PtTe$_2$-Pty possess a significantly lower value of $\Delta G_{H^*}$ as compared to the Pt2 site. To experimentally verify the gradually increased $\Delta G_{H^*}$ of Pt sites in PtTe$_2$, CV measurements of PtTe$_2$ NS, PtTe$_2$-200 NS, PtTe$_2$-600 NS, and Pt/C were performed in Ar purged 1.0 M KOH. As shown in Supplementary Fig. 27, the underpotentially deposited hydrogen (H$_{upd}$) peak of PtTe$_2$-600 NSs at ~0.21 V vs. RHE is negatively shifted relative to that of PtTe$_2$ NSs (~0.26 V vs. RHE) and PtTe$_2$-200 NSs (~0.25 V vs. RHE), suggesting weaker adsorption of hydrogen on PtTe$_2$-600 NSs[35,36]. Additionally, the H$_{upd}$ peak of PtTe$_2$-600 NSs is more negative than that of the Pt/C catalyst. From these mutually corroborating data (STM, $E_f$, $\Delta G_{H^*}$, and CV), it can be concluded that during heat treatment, the Te-SAVs in atomically thin PtTe$_2$ NSs migrate to form ordered Te-SAV clusters, which effectively reduces the adsorption strength of hydrogen on the Pt sites and thus drastically enhances the hydrogen evolution performance.

Last but not the least, the Fermi softness, defined by Eq. (1), was established to highlight the underlying origin between $\Delta G_{H^*}$ and the electronic structure of PtTe$_2$.

$$S_F = \int g(E)w(E)dE \tag{1}$$

Herein, $g(E)$ and $w(E)$ denote the density of states (DOS) and weight function, respectively[37], $w(E)$ is assigned by the derivative of Fermi-Dirac distribution function at non-zero temperature. As revealed in Fig. 4c, by including all Pt sites in different PtTe$_2$-xTe, an obvious linear relationship between $\Delta G_{H^*}$ and $S_F$ can be established. It is noteworthy that the PtTe$_2$-1Te, bearing a smaller

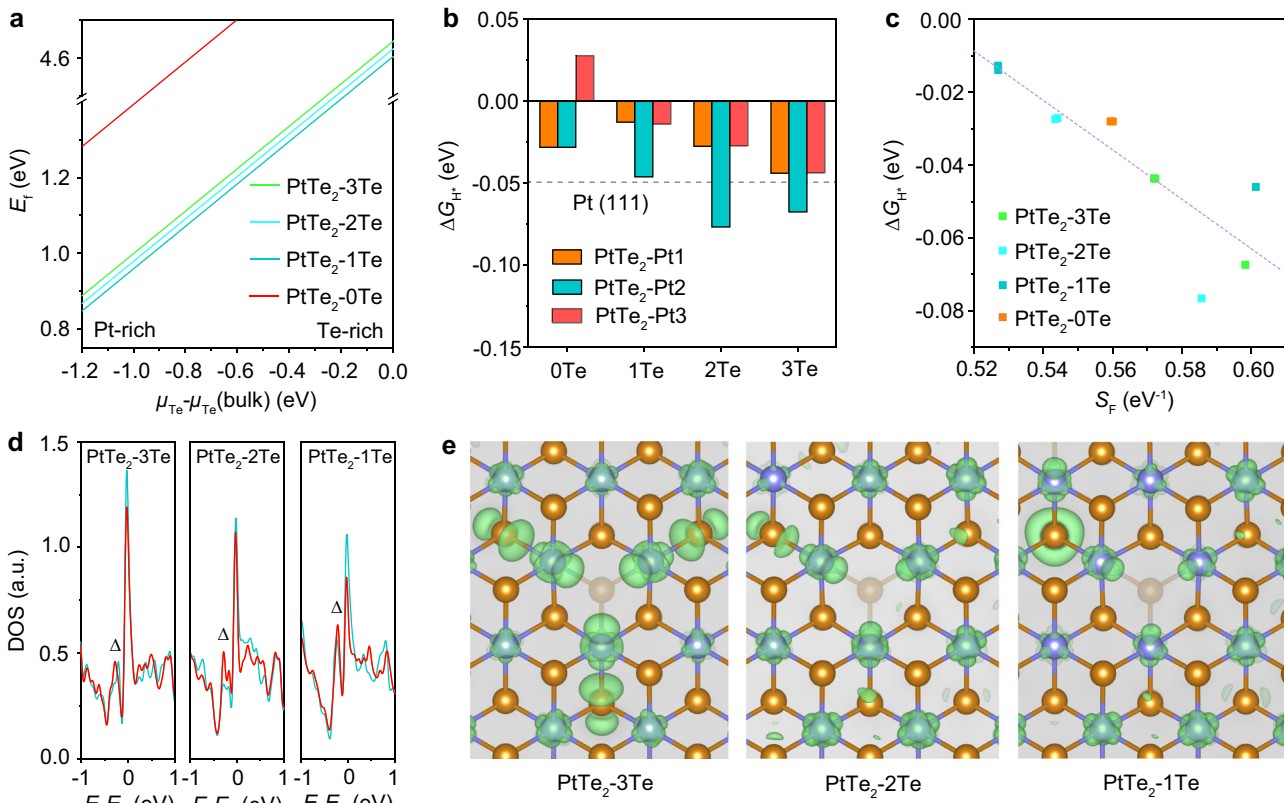

**Fig. 4 Electronic structure and binding strength of H. a** Formation energy of bilayer PtTe$_2$-3Te, PtTe$_2$-2Te, PtTe$_2$-1Te, and PtTe$_2$-0Te structures. **b** The calculated hydrogen adsorption free energy of different undercoordinated Pt sites in PtTe$_2$. **c** Dependence of hydrogen adsorption free energy on Fermi softness of Pt sites in various PtTe$_2$ structures. The nominal electronic temperature in weight function was set to 0.1. **d** DOS curves of Pt sites in PtTe$_2$-3Te, PtTe$_2$-2Te, and PtTe$_2$-1Te structures. The orange, blue, and red lines represent the DOS of Pt1, Pt2, and Pt3 sites in the corresponding PtTe$_2$-XTe structure, respectively. **e** The corresponding partial charge. The isosurface level was set to 0.002 e Å$^{-3}$.

$S_F$, features reduced hydrogen adsorption, suggesting that the DOS near the Fermi energy ($E_F$) of Pt sites greatly influences the $\Delta G_{H*}$. The changes in the DOS of Pt sites were further validated in Fig. 4d and Supplementary Fig. 28, where the peak of DOS near the $E_F$ displays a decreased order: PtTe$_2$-3Te > PtTe$_2$-2Te > PtTe$_2$-1Te, indicating a decreased DOS of Pt sites after the clustering of regular Te-SAVs. In addition, the total DOS of the Pt and Te atoms around the Te defects in PtTe$_2$-3Te, PtTe$_2$-2Te, and PtTe$_2$-1Te structure (Supplementary Fig. 29a-b) shows a similar variation trend. To verify this theoretical finding, we performed scanning tunneling spectroscopy (STS, d$I$/d$V$ vs. V) to probe the local DOS over different vacancy structures. The results show that near-Fermi local DOS of defect-free site, Te-SAV and trigonal Te-SAV in PtTe$_2$ (marked to red, orange, and blue symbols, respectively, in Fig. 2d) gradually decrease (Supplementary Fig. 29c). The reduced DOS of Pt sites near $E_F$ may result from two possible factors: (1) Electron transfer from Pt to Te atoms. As demonstrated in the in situ XANES results, the normalized Pt L$_3$-edge XANES spectra display a slight positive shift, while the Te absorption edge of PtTe$_2$-600 NSs is lower in energy, compared with that of the pristine PtTe$_2$ NSs. (2) The interaction between neighboring Te-SAVs as evidenced by the increased intensity of the DOS peak adjacent to the $E_F$ (noted with 'Δ' in Fig. 4d). In addition, partial charge density in the vicinity of $E_F$ was calculated to better visualize the decreased electronic states of Pt sites (Fig. 4e and Supplementary Fig. 30). The results confirm that all undercoordinated Pt sites possess interacting orbital directing to the center of Te vacancy, and the orbital volume gradually decreases as the neighboring Te-SAVs approach to each other. In conclusion, as both reduced DOS and deceased interacting orbital

volume of Pt sites are detrimental to hydrogen adsorption, the ordered clustering of Te-SAVs in atomically thin PtTe$_2$ NSs effectively reduces hydrogen adsorption, rendering them highly active in HER.

## Discussion

In conclusion, we have developed a facile exfoliation followed by thermal annealing approach to engineer the defects in atomically thin PtTe$_2$ NSs as a model catalyst to understand the correlation between electronic structure, adsorption energy, and hydrogen evolution activity of atomically defined Pt sites. The Te-SAVs in the exfoliated PtTe$_2$ NSs migrate to form ordered trigonal Te-SAV clusters during heat treatment, which effectively reduces the adsorption energy of hydrogen and promotes the kinetics of HER. In addition, benefiting from the exposed undercoordinated Pt sites with robust stability in atomically thin PtTe$_2$, the resulting catalysts exhibit superior activity and stability for HER, compared with commercial Pt/C catalysts. The finding here provides a strategy to engineer geometrically well-defined active sites via the clustering of atomic defects, which allows for the understanding of the correlations between electronic structure of catalytic center and catalytic performance.

## Method

**Materials**. Platinum (99.98%), tellurium (99.999%) were purchased from Alfa-Aesar and stored in the glove box. Tetrabutylammonium tetrafluoroborate (TBAB), dimethyl sulfoxide (DMSO), potassium hydroxide (KOH), and Nafion solution (5 wt.%) were purchased from Sigma–Aldrich, Pt/C (20 wt.%) catalysts were purchased from Alfa Aesar, and pure Argon gas (Purity: 99.999 %), pure hydrogen gas (Purity: 99.999 %) were purchased from Chem-Gas Pte Ltd (Singapore). All

chemical reagents were utilized as received without further purification. Water was purified through a Millipore system.

**Synthesis of bulk PtTe$_2$ crystals**. Bulk PtTe$_2$ crystals were synthesized via a CVT method in a three-zone tube furnace (OTF-1200X-III-S-UL, MTI Corporation, USA). First, 1 g of high-purity Pt (0.433 g) and Te (0.567 g) powders with a stoichiometric molar ratio of 1: 2 were ground thoroughly in a glove box. Then, the mixed powders were sealed in an evacuated quartz tube (length: 10 cm; external diameter: 13 mm; and wall thickness: 1 mm) under vacuum using an oxygen/hydrogen welding torch. Next, the sealed tube was placed in the furnace and heated to 1000 °C for 48 h. Afterwards, the temperature was further increased to 1150 °C for another 1 h. Finally, the furnace was slowly cooled to ambient temperature, and the bulk PtTe$_2$ crystals were collected.

**Electrochemical exfoliation of bulk PtTe$_2$ crystals**. Bulk PtTe$_2$ crystals were electrochemically exfoliated into atomically thin PtTe$_2$ NSs using an electrochemical workstation (CH Instruments, Inc., USA) consisting of a three-electrode system. Before exfoliation, the bulk PtTe$_2$ crystals were sliced into thin specimens and served as the working cathode, a Pt wire electrode was used as the counter electrode, and 0.05 M TBAB-DMSO solution was used as the electrolyte. The exfoliation of PtTe$_2$ crystals was performed by applying a bias of −5.0 V on the working electrode. After exfoliation, the obtained product was collected by centrifugation, washed with plenty of H$_2$O and ethanol, and dried in a vacuum oven.

**Thermal treatment of PtTe$_2$ NSs**. The exfoliated PtTe$_2$ NSs were placed in a quartz boat and heated in a tube furnace to T (T = 200, 400, or 600 °C) at 5 °C min$^{-1}$ in argon gas (flowrate: 200 sccm) for 1 h. After cooling to room temperature, the PtTe$_2$-T NSs were obtained.

**Electrochemical measurements**. The electrochemical measurements were performed on a standard three-electrode system using a rotating disk electrode setup (WaveVortex 10, USA) at room temperature (22 °C). A catalyst coated glassy carbon electrode (geometric area: 0.19625 cm$^2$), a carbon rod and a Hg/HgCl electrode (KCl-saturated) were used as the working, counter, and reference electrode, respectively. Argon saturated 1.0 M KOH or 0.5 M H$_2$SO$_4$ solution was used as the electrolyte. Before loading catalysts, glassy carbon electrode (GCE) was successively polished with 1.0, 0.3, and 0.05 mm Al$_2$O$_3$ slurry to obtain an ultra-clean surface. Afterwards, 4 mg of catalysts were dispersed in 1 ml of ethanol with 8 μl of Nafion 117 solution and sonicated for ~2 min to obtain a homogeneous catalyst ink. For commercial Pt/C (20 wt.%), 40 μl of the catalyst ink was dropped onto the GCE and allowed to dry at room temperature. Thus, the average loading of the catalyst is 0.809 mg cm$^{-2}$, and the total mass loading of Pt is 0.162 mg cm$^{-2}$. For PtTe$_2$ (n$_{Pt}$: n$_{Te}$ = 0.61), 16.6 μl of the catalyst ink was dropped onto the GCE and dried at room temperature. The average loading of the catalyst is 0.336 mg cm$^{-2}$, and the Pt content is ~0.162 mg cm$^{-2}$, which is the same as that in Pt/C (20 wt.%) catalysts. To investigate the influence of Pt/C (20 wt.%) catalyst loading amount on HER, the catalyst loading-dependent current density at a particular overpotential was studied. As shown in Supplementary Fig. 31, when the loading amount of Pt/C increases from 0.4 mg cm$^{-2}$ to 0.9 mg cm$^{-2}$, the current density almost increases linearly (blue dash line). At this stage, the number of active sites dominate the HER process, rather than other factors (e.g., mass transport, conductivity). At the loading amount of Pt/C of 1.0 mg cm$^{-2}$, the current density starts to decrease sharply.

All LSV curves were collected at a rotation speed of 1600 rpm and a sweep rate of 5.0 mV s$^{-1}$ with 80 % of ohmic drop compensation. The reference electrode was calibrated in a H$_2$-saturated 1.0 M KOH and 0.5 M H$_2$SO$_4$ solution at a scan rate of 5.0 mV s$^{-1}$ (Supplementary Fig. 32)[15,34,38]. EIS measurements were conducted at an overpotential of 100 mV in the frequency range of 0.01 to 10$^5$ Hz with an amplitude of 5 mV. The accelerated stability test was performed by measuring the LSV curves before and after 20,000 continuous CV cycles in the potential range from −0.2 to 0.0 V vs. RHE with a sweep rate of 200 mV s$^{-1}$. The durability of the catalyst was evaluated by the chronopotentiometry method.

**ECSA calculation**. The CV measurement was performed at potentials between 0 and 1.25 V vs. RHE at a scan rate of 50 mV s$^{-1}$ in Argon-purged 1.0 M KOH solution at room temperature. ECSA of PtTe$_2$ NSs, PtTe$_2$-200 NSs, PtTe$_2$-400 NSs, PtTe$_2$-600 NSs, and Pt/C (20 wt.%) were then calculated based on the reported method and Eq. (2)[28–30]:

$$ECSA = \frac{Q_H}{0.21 \times [Pt]} = \frac{S_{area}}{V_{scan} \times 0.21 \times [Pt]} \quad (2)$$

where $Q_H$ (mC) is the average charge integrated from hydrogen adsorption/desorption process (0.0–0.5 V vs. RHE) on the CV curve, which can be determined by the ratio of integrating hydrogen adsorption/desorption peak area with the subtraction of the double layer to scanning velocity (0.05 V s$^{-1}$). 0.21 mC cm$^{-2}$ is the electrical charge associated with monolayer adsorption of hydrogen on Pt, and [Pt] is the loading of Pt on the working electrode (0.0318 mg in our experiments).

**TOF calculation**. To calculate the TOF per undercoordinated Pt sites in the PtTe$_2$ catalyst, we used the following Eq. (3):

$$TOF = \frac{\# \text{ Total hydrogen turnover per geometric aera}}{\# \text{ Active sites per geometrica area}} \quad (3)$$

The total number of hydrogen turn overs was calculated from the current density according to the Eq. (4)[39,40]:

$$Total\,hydrogen\,turnovers = \left(J\frac{mA}{cm^2}\right)\left(\frac{1C/s}{1000mA}\right)\left(\frac{1mole^-}{96485.3C}\right)\left(\frac{1molH_2}{2mole^-}\right)\left(\frac{6.02 \times 10^{23}molecules\,H_2}{1molH_2}\right)$$
$$= 3.12 \times 10^{15}\,\frac{H_2/s}{cm^2}\,per\,\frac{mA}{cm^2} \quad (4)$$

The upper limit number of active sites was calculated based on the hypothesis that all defective Pt sites in the PtTe$_2$ catalysts formed active centers and all of them were accessible to the electrolyte. The real number of active and accessible undercoordinated Pt sites should be considerably lower than the calculated value. According to ICP–OES results that n$_{Pt}$: n$_{Te}$ is 0.61, thus, the percentage of defective Pt sites in all Pt sites was calculated to be 18%. Thus, the active sites per geometrical area were obtained according to the Eq. (5):

$$\#Active\,sites(Pt)$$
$$= \left(\frac{\text{defective Pt wt.\%} \times \text{catalyst loading}\left(\frac{g}{cm^2}\right)}{\text{Pt Mw}\left(\frac{g}{mol}\right)}\right)\left(\frac{6.022 \times 10^{23}}{1\,mol\,Pt}\right)$$
$$= \left(\frac{48.25\,\% \times 18\,\% \times 0.3357 \times 10^{-3}\left(\frac{g}{cm^2}\right)}{195.05}\right)\left(\frac{6.022 \times 10^{23}}{1}\right) \quad (5)$$
$$= 9 \times 10^{16}\,\text{sites cm}^{-2}$$

Finally, the current density from the LSV polarization curve can be converted into TOF values according to the Eq. (6):

$$TOF = \frac{3.12 \times 10^{15}}{9 \times 10^{16}} \times |J| = 0.035|J| \quad (6)$$

**STM/STS measurement**. The STM/STS measurements were conducted at T = 4.8 K in the CreaTec STM/AFM system under UHV. Before each measurement, the STM tip was calibrated on an Au(111) surface by checking the Shockley surface state. All the dI/dV spectra were taken through the standard lock-in technique with a modulation voltage of 10 mV and frequency of 713 Hz.

**Computational details**. DFT calculations were carried out using the plane-wave technique with exchange-correlation interactions modeled by GGA-PBE functional[41], as implemented in the Vienna ab initio Simulation package[42]. The ion–electron interactions were described by the projector augmented plane-wave approach[43], and the cutoff energy was set to 400 eV. Structural optimizations were performed by minimizing the forces on all atoms to below 0.02 eV Å$^{-1}$ and the energy to below 10$^{-5}$ eV. Monkhorst-Pack method was adopted to sample the k-space with 7 × 7 × 5 mesh for PtTe$_2$ unit-cell and 2 × 2 × 1 mesh for the surface of bilayer, respectively. The van der Waals correction was included using the Becke-Jonson damping with function parameters of the D2 method by Grimme et al.[43]. After fully relaxed, the lattice parameters of PtTe$_2$ unit-cell were optimized to a = 4.01 Å and c = 5.00 Å, and the phonon spectrum (Supplementary Fig. 21) confirmed its dynamic stability in terms of no imaginary frequency. To explore the catalytic performance of layered PtTe$_2$, a (8 × 8) slab with two layers along the (001) axis was constructed accompanying with a vacuum layer of 15 Å to avoid the interaction between neighboring images.

The free energy analysis method developed by Nørskov et al.[44], was used to predict the reaction activity and the computational details can be referred to previous report[32]. We adopted Eq. (2): $\Delta G_{H^*} = \Delta E_{H^*} + 0.24$ eV (where $\Delta E_{H^*}$ is the adsorption energy of H atom) to evaluate the adsorption free energy of H on different sites of PtTe$_2$ surface. The formation energy of defective PtTe$_2$ was calculated by the following Eq. (7):

$$E_f = E^t(def) - E^t(ideal) + \sum N_{Te}\mu_{Te} \quad (7)$$

where $E^t(def)$ and $E^t(ideal)$ represent the total energy of defective and pristine PtTe$_2$, and $\mu_{Te}$ and $N_{Te}$ represent the chemical potential of Te in PtTe$_2$ and the vacancy number, respectively. Since it is impossible to obtain an accurate value of $\mu_{Te}$, it varies between two limits, namely the $\mu_{Te(bulk)}$ and $1/2(E_{PtTe_2} - \mu_{Pt(bulk)})$.

## Data availability

The data supporting this study are available from the corresponding author upon reasonable request.

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

## Acknowledgements

This work was financially supported by the National Natural Science Foundation of China (21972094 and 22005244), Guangdong Special Support Program, Pengcheng Scholar program, Shenzhen Peacock Plan (KQJSCX20170727100802505 andKQTD2016053112042971), Natural Science Foundation of Ningbo City (202003N4052), the Ministry of Education of Singapore (AcRF Tier 1 M4011021.120, 2015-T1-002-108, R-143-000-A75-114, R-143-000-B47-114, Tier 1: RG4/20 and Tier 2: MOET2EP10120-0002), and Agency for Science, Technology and Research (A*Star IRG: A20E5c0080). The computational work for this article was partially performed on resources of the National Supercomputing Centre, Singapore (https://www.nscc.sg).

## Author contributions

X.L., B.L., J.L., and C.S. conceived the research. X.L. and Y.F. carried out the synthesis and performed materials characterization and electrochemical measurements. J.W. conducted the theoretical calculation. H.F. performed the STM measurement. S.X. conducted the XAFS measurements. X.Z. conducted the HAADF-STEM measurement. H.X., W.Y., D.X., and X.H. assisted in the synthesis and characterization of materials. X.L., Y.F., W.J., J.L., and B.L. wrote the manuscript. All authors discussed the results and commented on the manuscript.

## Competing interests

The authors declare no competing interests.
