## [Peer Review File · Nature Communications]

REVIEWER COMMENTS

Reviewer #1 (Remarks to the Author):

Li and coworkers' manuscript presents a nice research work regarding the ordered clustering of single atomic Te vacancies in atomically thin PtTe₂ for electrocatalytic application. In this manuscript, the authors first prepare atomically thin 2D PtTe₂ nanosheets with well-dispersed single atomic Te vacancies by electrochemically exfoliating bulk PtTe₂ crystals. Then, heat treatment induces the migration of the random Te-SAVs to form ordered Te-SAV clusters. This discovery is interesting and crucial because accurately tailoring the vacancy structure at the atomic level to improve the intrinsic catalytic activity is generally quite difficult. When used as electrocatalysts for HER, the heat-treated PtTe₂ materials with Te-SAV clusters reveal significantly improved catalytic activity, much better than that of the state-of-the-art Pt/C catalyst and other reported HER electrocatalysts. Notably, the results of the experiment and calculation are mutually verified. Finally, in the current version, several concerns raised by the reviewers have been well-addressed. Thus, publishing this work on Nature Communications is recommended after addressing the following concerns.

- (1) What is the purpose of presenting C mapping in Fig. 1g? If this figure shows that TBAB or its decomposition products on the surface of exfoliated PtTe₂ are removed, the authors should give the corresponding description in the main text.
- (2) Descriptions and evidence regarding single vacancies are Te vacancies rather than Pt vacancies need to be added into the manuscript because this point is vital for the whole work.
- (3) For the HER application, PtTe₂ NSs show substantially larger geometric HER current density than bulk PtTe₂ crystals at the same overpotential. The author claims that the enhanced activity comes from the more exposed active sites of PtTe₂ NSs after exfoliation. The authors need to cite related references to support this conclusion.
- (4) Defects engineering is a common strategy to improve the catalytic activity of nanomaterials. Previously published work on defect engineering in TMDs materials usually focuses on improving the number of vacancies and doping vacancies via heteroatoms. Accurately tailoring vacancy structure in TMDs materials at the atomic level to improve the intrinsic is seldomly reported. The authors can further highlight this point in the main text.
- (5) In Fig. S 28, the calculated DOS shows a peak at Fermi level, but the measured dI/dV spectrum shows a dip at Fermi level. The author should explain the difference.
- (6) In the calculation part, what is the interacting orbital volume of Pt sites? The authors should give the corresponding description and explain the functions of the orbital volume of Pt sites. The authors also used the Fermi softness to relate Pt sites' electronic structure to the intrinsic activity, rather than the commonly used d-band center. Are there any other related references or evidence regarding the inadaptation of the d-band center?
- (7) Transition metal phosphides have recently identified as high-performance HER catalysts with Pt-like activity (J. Am. Chem. Soc. 2014, 136, 7587–7590; Adv. Mater. 2017, 29, 1602441; Adv. Energy Mater. 2017, 7, 1700020), which could be discussed in the Introduction section.

Reviewer #2 (Remarks to the Author):

This manuscript reports the preparation of atomically thin two-dimensional PtTe₂ nanosheets having well dispersed single atomic Te vacancies (Te-SAVs) and Pt sites as a catalyst. The Te-SAVs were created by electrochemical exfoliating bulk PtTe₂ crystals to expose the undercoordinated and stabilized Pt sites. The PtTe₂ nanosheets with ordered clusters of Te-SAVs exhibit hydrogen evolution reaction (HER) activity at low overpotential and Tafel slope.

PtTe based catalyst has already been reported in Nano Energy for HER (Nano Energy 2019, 61, 346–351347), and moreover, the catalytic activity of the given catalyst is not the best among the

reported noble metal-based catalysts. In brief, this study lacks the desired originality both in catalyst design and activity, so the manuscript is not suitable for publication in Nature Communications.

1. It seems the occurrence of vacancies are significantly less in the sheet. It means the number of defects sites are relatively sparse to influence the materials' overall activity. This situation overshadows the theory the authors have built up to explain the enhanced activity. The activity could be just because of the enhanced exposed surface area.

2. The LSV curves of the samples in 1.0 M KOH (Fig. 3) clearly indicates the enhancement of HER performance with respect to temperature. It is a clear indication of enhanced crystallinity and conductivity, which enhances the activity. The improved activity is not related to defect sites; instead, it is related to crystallinity and conductivity (As the crystallinity is enhanced after annealing at 600 °C), as Pt is a well known the most active HER catalyst.

3. The straightforward comparison between Pt/C (20%) with PtTe₂ will not be fair, as the amount of Pt in PtTe₂ is much higher than the Pt/C. It means the per unit mass activity of Pt/C is much better than the PtTe₂.

4. Moreover, it is not the best-reported values compared to literature available catalysts. There are many Ru and Ir based catalysts, which are better than this catalyst. The authors have only shown the catalysts which are less active their catalyst.

Reviewer #3 (Remarks to the Author):

My previous comments have been fully addressed. I strongly support the publication of this work.

Point-by-point response letter

We thank the reviewers for their valuable comments (*words in italics*), response to which will undoubtedly improve our manuscript and clarify issues we may have failed to point out in the first version. We have provided our point-by-point response to each comment (in blue). The revised part for the Main Text and Supplementary Information are marked with blue.

Reviewer #1 (Remarks to the Author):

Li and coworkers' manuscript presents a nice research work regarding the ordered clustering of single atomic Te vacancies in atomically thin PtTe₂ for electrocatalytic application. In this manuscript, the authors first prepare atomically thin 2D PtTe₂ nanosheets with well-dispersed single atomic Te vacancies by electrochemically exfoliating bulk PtTe₂ crystals. Then, heat treatment induces the migration of the random Te-SAVs to form ordered Te-SAV clusters. This discovery is interesting and crucial because accurately tailoring the vacancy structure at the atomic level to improve the intrinsic catalytic activity is generally quite difficult. When used as electrocatalysts for HER, the heat-treated PtTe₂ materials with Te-SAV clusters reveal significantly improved catalytic activity, much better than that of the state-of-the-art Pt/C catalyst and other reported HER electrocatalysts. Notably, the results of the experiment and calculation are mutually verified. Finally, in the current version, several concerns raised by the reviewers have been well-addressed. Thus, publishing this work on Nature Communications is recommended after addressing the following concerns.

(1) What is the purpose of presenting C mapping in Fig. 1g? If this figure shows that TBAB or its decomposition products on the surface of exfoliated PtTe₂ are removed, the authors should give the corresponding description in the main text.

Response: We thank the reviewer for the positive comments and helpful suggestion. According to STEM elemental mapping results displayed in Fig. 1g, we can see that only C signal from the carbon-coated copper grid is detected after long-time signal collection. No signal of C is detected on the surface of exfoliated PtTe₂. The purpose of presenting C mapping in Fig. 1g is to show that TBAB and/or its decomposition products on the surface of exfoliated PtTe₂ are removed. According to the reviewer's suggestion, the related descriptions have been added in the revised manuscript and marked as blue.

(2) Descriptions and evidence regarding single vacancies are Te vacancies rather than Pt vacancies need to be added into the manuscript because this point is vital for the whole work.

Response: We thank the reviewer for the comment. The evidences regarding single vacancies are Te vacancies rather than Pt vacancies come from the following aspects: (1) The molar ratio

of Pt : Te in the prepared PtTe₂ catalysts (0.61) is much larger than the theoretical stoichiometric ratio of PtTe₂ (0.5) according to the ICP-OES results (Supplementary Table 1), from which the atomic percentage of Te vacancies in the prepared PtTe₂ catalysts is calculated to be 18 at.%. (2) Vacancies can be directly visualized with distinguishable contrast in the few-layer PtTe₂ NSs at the lower left corner of Fig. 1f. The corresponding line intensity profile (inset in the upper portion of Fig. 1f) combined with the atomic model (inset at the lower right corner of Fig. 1f) indicates that the point defect corresponds to Te vacancy. (3) The observed atoms in the atom-resolved STM images (Fig. 2b) are located at the topmost layer of PtTe₂ with a sandwiched Te-Pt-Te structure, which has been demonstrated in other transition metal dichalcogenide systems, such as MoS₂¹, MoSe₂², MoTe₂³, WSe₂⁴, and PtSe₂⁵. By superimposing the atomic structure over the lattice, it can be found that all the black dots are Te sites rather than Pt sites (Fig. 2b). (4) Defects of metal and chalcogen show completely different appearances in the STM images according to the reported First-Principles calculations⁶. For example, both DFT calculations and experiments have demonstrated that Se vacancy in PtSe₂ is a dip site surrounded by six protrusion sites. In contrast, Pt vacancy displays three protrusions without a dip at the Se sites in the neighbor⁵. By comparing STM results in PtSe₂, we can see that the black dots observed in our experiments are chalcogen vacancies (Te vacancies) rather than Pt vacancies. According to the reviewer's suggestion, the related descriptions have been added in the revised manuscript and marked as blue.

(3) For the HER application, PtTe₂ NSs show substantially larger geometric HER current density than bulk PtTe₂ crystals at the same overpotential. The author claims that the enhanced activity comes from the more exposed active sites of PtTe₂ NSs after exfoliation. The authors need to cite related references to support this conclusion.

Response: We thank the reviewer for the helpful suggestion. The bulk PtTe₂ crystals synthesized by chemical vapor transport technique have closely stacked lamellar architecture with the interlamellar spacing of 3.52 Å (Supplementary Figs. 2 and 3), thus, only a small amount of active sites can be used for the catalytic reaction. After electrochemical exfoliation (detailed in Supplementary Figs. 3 and 4), the bulk PtTe₂ crystals are exfoliated into the atomically thin two-dimensional PtTe₂ nanosheets with fully exposed undercoordinated Pt sites

(Fig. 1d). As a result, PtTe₂ NSs show substantially larger geometric HER current density than bulk PtTe₂ crystals at the same overpotential. Similar results are also reported in other materials, such as NiPS₃⁷ and Sb⁸. According to the reviewer's suggestion, the related references to support this conclusion have been added in the revised manuscript.

(4) Defects engineering is a common strategy to improve the catalytic activity of nanomaterials. Previously published work on defect engineering in TMDs materials usually focuses on improving the number of vacancies and doping vacancies via heteroatoms. Accurately tailoring vacancy structure in TMDs materials at the atomic level to improve the intrinsic is seldomly reported. The authors can further highlight this point in the main text.

Response: We thank the reviewer for the helpful suggestion. Vacancy engineering is a vital strategy to modulate the surface electronic structure of electrocatalysts to improve their catalytic activities⁹⁻¹². Many electrocatalysts with different vacancies have been prepared and applied in various electrocatalytic reactions⁹⁻²⁴. Nevertheless, the diversity and complexity of vacancies in these catalysts prevent the in-depth understanding of the vacancy–catalysis relationship. As a result, there are still many open questions that have not been answered in this field. For example, how to precisely create atomic vacancies with a well-defined structure? How to rationally regulate vacancy structure to improve the intrinsic electrocatalytic activity? What is the underlying correlation between vacancy and catalytic activity? and etc.

Many synthetic methods, such as controlled growth^{13, 14}, etching^{15-19, 24}, heat and plasma treatment²⁰⁻²², and template synthesis²³, have been applied to prepare electrocatalysts with vacancies. However, up to now, it is still challenging to accurately regulate specific vacancies with atomic precision in materials as the conventional synthesis methods typically have no selectivity to particular sites^{9, 16, 19, 25}, making the study of vacancy-catalysis relationship very challenging. Furthermore, the vacancies in the materials are usually randomly dispersed without forming a well-defined structure¹³. Different from traditional works, our work shows that a large amount of uniformly dispersed single atomic Te vacancies can be created in PtTe₂ crystals by chemical vapor transport method. Besides, these uniformly dispersed Te vacancies with the well-defined structure can be fully exposed after electrochemically exfoliating the bulk PtTe₂ crystals. More importantly, previous works on vacancy engineering mainly focus on

regulating the number of vacancies as well as doping vacancies via heteroatoms^{17, 18, 20, 25-29}. Accurately tailoring vacancy structure at the atomic level to improve the intrinsic catalytic activity remains a grand challenge. Our work shows that single atomic vacancies can migrate, and form ordered atomic vacancy clusters with the trigonal structure, which effectively regulates the bindings between the reaction intermediates and catalytic sites and improves the intrinsic catalytic activity. According to the reviewer's suggestion, the related descriptions have been added in the revised manuscript and marked as blue.

(5) In Fig. S28, the calculated DOS shows a peak at Fermi level, but the measured dI/dV spectrum shows a dip at Fermi level. The author should explain the difference.

Response: We thank the reviewer for raising the question. First of all, although the STS result shows a dip at the Fermi level, it still displays a reduction of density of states of undercoordinated Pt sites around the Fermi level from bulk PtTe₂ to single Te vacancy and to Te-SAV clusters. Secondly, the STS result has similar structure with the calculated DOS. The STS result shows a peak at -0.117 eV and a dip at -0.01 eV while the calculated result shows a peak at -0.025 eV and a dip at 0.087 eV. Such difference is more likely to be a result from Fermi level shifting. In 2D materials, intrinsic vacancies are one typical resource for electron or hole doping. For example, sulfur vacancies introduce n-doping in MoS₂ and phosphorus vacancies introduce p-doping in black phosphorus^{30, 31}. Hence, the vacancy density will affect the doping level in the material and shift the Fermi level. The calculated results are based on an 8 × 8 supercell with 3 Te vacancies. The vacancy density in such system is around 9.0 nm⁻² while the statistical result for the vacancy density is 0.3 nm⁻² from STM. The different vacancy density will result in a shift of Fermi level. Then the peak from calculated DOS will shift to lower energy and leave a dip at the Fermi level in STS measurement.

(6) In the calculation part, what is the interacting orbital volume of Pt sites? The authors should give the corresponding description and explain the functions of the orbital volume of Pt sites. The authors also used the Fermi softness to relate Pt sites' electronic structure to the intrinsic activity, rather than the commonly used d-band center. Are there any other related references or evidence regarding the inadaptation of the d-band center?

Response: We appreciate the reviewer for the insightful comments and suggestions. According to frontier orbital (FO) theory, the interacting intensity between orbitals of catalyst and adsorbent can be determined by three factors, namely accessible energy, matchable symmetry and sufficient overlapping. If the energy and symmetry maintain the same, the overlapping of two interacting orbitals from catalyst and adsorbent will become a decisive factor to impact their interacting intensity. In this work, the interacting orbitals of Pt atoms are represented by partial electron density around the Fermi level of PtTe₂ system³². A larger iso-surface (i.e. the larger volume enclosed by the surface) indicates a higher electron density around the corresponding atoms when the iso-value is fixed, thus can give rise to more effective overlapping with the electron density (i.e. the FO orbital) of adsorbent with stronger adsorption. In our work, the adsorption of H on Pt atoms in defective PtTe₂ gradually decreases when the neighboring Te-SAVs approach to each other owing to their gradually decreased orbital volume. According to the reviewer's suggestion, the related descriptions have been added in the revised manuscript and marked as blue.

Although the d-band center theory has been commonly used to predict reactivity of metal or cluster systems, its adaptation in single-atom and compound system is still dubious. The distinction may come from different orbital contribution of metal atoms (adsorbing site) in different systems³³⁻³⁵. Actually, this descriptor can even produce contradictory results for some metal alloys with complicated structures³⁶. In general, for most metal (or simple surface alloy) systems, the d-orbitals of metal atoms will hybridize with each other strongly. It also can simultaneously participate in the interaction with orbitals of adsorbent. Thus, an average weight of d-band (d-band center) can be useful to evaluate the reactivity of metal atoms³⁷. However, for substrate supported single-atom system, the orbital distribution will be affected by lattice-field of substrate, which usually results in the slight hybridization with each other. As a result, the orbital overlapping between metal atom and adsorbent has to consider their wave function matching, and always only part of d-orbitals show dominant role in the interaction, under which the d-band center may not be suitable any more³³. The compound containing M-X covalent bond also follows similar principle as single-atom system, since d-orbitals with different spatial directions show distinctive contribution to the interaction strength.

(7) *Transition metal phosphides have recently identified as high-performance HER catalysts with Pt-like activity (J. Am. Chem. Soc. 2014, 136, 7587–7590; Adv. Mater. 2017, 29, 1602441; Adv. Energy Mater. 2017, 7, 1700020), which could be discussed in the Introduction section.*

Response: We thank the reviewer for the helpful suggestion. Pt is generally recognized as the state-of-the-art HER catalyst³⁸. However, the limited storage and very high price of Pt impede its large-scale industrial application. Thus, it is highly worthwhile to develop non-noble metal based electrocatalysts with Pt-like activity for HER. Recently, several transition metal phosphides (TMPs) have been proven as highly active and inexpensive catalysts for HER, such as CoP³⁹⁻⁴¹, Ni₂P⁴²⁻⁴⁴, MoP^{45, 46}, RuP₂⁴⁷, Cu₃P⁴⁸, FeP⁴⁹, and Ni₅P₄⁵⁰. The corresponding investigations have revealed that P atoms with electronegativity usually can draw electrons from metal atoms in the TMPs. Thus, the negatively charged P can act as a base to trap positively charged proton during HER⁵¹. Furthermore, increasing the atomic percentage of P in the same metal phosphide can also effectively enhance the HER activity⁵². According to the reviewer's suggestion, the related descriptions about transition metal phosphides have been added in the introduction section and marked as blue.

Reviewer #2 (Remarks to the Author):

This manuscript reports the preparation of atomically thin two-dimensional PtTe₂ nanosheets having well dispersed single atomic Te vacancies (Te-SAVs) and Pt sites as a catalyst. The Te-SAVs were created by electrochemical exfoliating bulk PtTe₂ crystals to expose the undercoordinated and stabilized Pt sites. The PtTe₂ nanosheets with ordered clusters of Te-SAVs exhibit hydrogen evolution reaction (HER) activity at low overpotential and Tafel slope. PtTe based catalyst has already been reported in Nano Energy for HER (Nano Energy 2019, 61, 346–351347), and moreover, the catalytic activity of the given catalyst is not the best among the reported noble metal-based catalysts. In brief, this study lacks the desired originality both in catalyst design and activity, so the manuscript is not suitable for publication in Nature Communications.

Response: We appreciate the reviewer for the helpful comments on our manuscript. Herein we would like to highlight the novelty of this work again from the aspects of catalyst design and catalytic activity.

We agree with the reviewer that PtTe based catalysts have been studied as electrocatalysts for HER over recent years⁵³⁻⁵⁵. Taking the published work mentioned by the reviewer as an example, Liu et al. reported a solid-liquid phase chemical strategy for *in situ* formation of cation vacancies in a five-fold twinned PtPdRuTe anisotropic structure (v-Pd₃Pt₂₉Ru₆₂Te₆ AS)⁵⁴. However, as stated in this paper, the v-Pd₃Pt₂₉Ru₆₂Te₆ AS consists of PtPdRuTe alloy and a small amount of PtTe₂, rather than the pure PtTe₂ material, which is demonstrated by their XRD result as shown in Fig. S1. Thus, the atomic structure of this material is very complex, making it difficult as a model catalyst to investigate the structure-property relationship and study the catalytic mechanism at the atomic scale. Completely different from the traditional PtTe-based materials, Pt and Te atoms in PtTe₂ in our work crystallize in an atomically precise CdI₂-type trigona (1 T) structure (P $\bar{3}m1$, a = b = 4.03 Å, c = 5.33 Å, Supplementary Fig. 1)⁵⁶, rather than are randomly distributed, which have been demonstrated by the XRD in Fig. 1c, high-resolution HAADF-STEM image and atomic model in Fig. 1f. Consequently, this atomically-thin 2D PtTe₂ material offers an ideal model catalyst for providing atomic-level insights into the Pt active site–catalytic performance relationship due to its well-defined atomic structure.

For the aspect of catalytic performance, we agree that our PtTe₂ catalyst's HER performance is not the best in the literature. However, as shown in Fig. 3e and Supplementary Table S2, our PtTe₂ catalyst still outperforms many of the reported highly active HER catalysts, especially the Pt-based HER catalysts. It is commonly known that many factors can affect the activity measurement of HER, such as catalyst loading amount⁵⁷⁻⁶², electrode preparation procedure, iR correction⁶³⁻⁶⁶, reference electrode calibration^{60, 61, 67-70}, temperature⁷¹⁻⁷⁴, scan speed⁷⁵, and so on. However, for example, many published papers with very high HER activities even do not contain any information about reference electrode calibration, as marked in the Supplementary Table S2. In addition, some published work with very high HER activity even selected Pt wire as the counter electrode⁷⁶, which is obviously a mistake because it has been reported that the Pt electrode can be dissolved from the oxidation reaction (Pt⁰ to Pt²⁺) and the dissolved Pt²⁺ can transfer to the working electrode surface⁷⁷⁻⁷⁹. Apart from the activity, catalytic stability is equally important. Accelerated cyclic voltammetry (CV) test elucidates the robustness of PtTe₂-600 NSs in HER catalysis with virtually unchanged polarization curves even after 20,000 CV cycles (Fig. 3h). Furthermore, long-term chronopotentiometry measurement shows a slight change of overpotential after 24 h at a constant current density of 200 mA cm⁻² for PtTe₂-600 NSs, much better than that of Pt/C catalyst (Fig. 3i).

The novelty of our work is that we designed and prepared atomically thin 2D PtTe₂ nanosheets with exposed and stabilized undercoordinated Pt sites as an atomically controllable Pt-based model catalyst to understand the correlation between electronic structure, adsorption energy, and catalytic property of atomic Pt sites. Also, we found that thermal treatment could drive migration of the well-dispersed single atomic Te vacancies in PtTe₂ to form thermodynamically stabilized, ordered trigonal well-dispersed single atomic Te vacancy clusters. This finding provides a new strategy to engineer geometrically well-defined active sites via clustering of atomic defects.

Fig. S1 XRD pattern of $v\text{-Pd}_3\text{Pt}_{29}\text{Ru}_{62}\text{Te}_6$ AS⁵⁴.

1. It seems the occurrence of vacancies are significantly less in the sheet. It means the number of defects sites are relatively sparse to influence the materials' overall activity. This situation overshadows the theory the authors have built up to explain the enhanced activity. The activity could be just because of the enhanced exposed surface area.

Response: We thank the reviewer for the comments. According to the ICP-OES results in Supplementary Table S1, the $n_{\text{Pt}} : n_{\text{Te}}$ in PtTe_2 nanosheets is about 0.61, much larger than the nominal molar ratio in PtTe_2 ($n_{\text{Pt}} : n_{\text{Te}} = 0.5$). Thus, the atomic percentage of Te vacancies is calculated to be 18 at.%.

On one hand, compared with other reported transition metal dichalcogenide (TMDs) materials, such as classical 2D MoS_2 materials where the optimal vacancy concentration is usually much less than 18 at.%^{14, 80-82}, the occurrence of Te vacancies (18 at.%) in the PtTe_2 sheets obviously is not less. For example, Meng et al. employed laser ablation to produce 8 at.% S-vacancies in the basal plane of MoS_2 nanosheets, thus decreasing the overpotential at the current density of 10 mA cm^{-2} from 260 mV to 182 mV⁸⁰. In addition, Tsai et al. used an electrochemical desulfurization route to create S-vacancies in MoS_2 with various concentrations up to 7.5 at.%, and the optimal concentration of 6.25 at.% decreased the overpotential in half^{14, 81}. Recently, Wang et al. theoretically predicted and experimentally demonstrated that S-vacancy state (concentration and distribution) in MoS_2 could significantly influence the HER performance¹⁴. When the S vacancy concentration reaches 12.50 at.%, the

defective MoS₂ exhibits the optimal HER performance. Similar results are further demonstrated in other defective TMD materials, such as MoSe₂⁸³. Based on these reports, such a high amount of Te vacancies (18 at.%) in PtTe₂ is enough to significantly influence the material's overall catalytic activity.

On another hand, catalysis is a surface process, in which the coordinatively unsaturated metal sites can serve as the active sites in catalysis. Hence, the number of unsaturated atoms plays a crucial role in influencing the catalytic properties^{9, 84-89}. Usually, increasing the number of coordinatively unsaturated atoms provides an efficient strategy to promote the catalytic activity^{84, 85, 88}. For example, Liu's group demonstrated that surface reactivity of transition metal oxides increases monotonically with the density of coordinatively unsaturated metal cation⁸⁴. In our PtTe₂ materials, the presence of 18 at.% Te defects can help to create 8.7 wt.% coordinatively unsaturated Pt atoms near these Te defects. Compared with the reported highly active Pt-based single atom catalysts for HER, where the weight percentage of coordinatively unsaturated Pt atoms in these catalysts is usually less than 3 wt.%, such as Pt₁/N-C (2.5 wt.%)⁹⁰, Pt₁/OLCs (0.27 wt.%)⁶⁸, Pt/np-Co_{0.85}Se (~1.03 wt.%)⁹¹, and Mo₂TiC₂Tx-Pt_{SA} (1.2 wt.%)⁹², the weight percentage of coordinatively unsaturated Pt atoms in PtTe₂ is high enough to impact the HER.

What's more, the bulk PtTe₂ crystals synthesized by chemical vapor transport technique have closely stacked lamellar architecture with interlamellar spacing of 3.52 Å (Supplementary Fig. 2 and 3). After electrochemical exfoliation (detailed in Supplementary Figs. 3 and 4), the bulk PtTe₂ crystals are exfoliated into atomically thin two-dimensional PtTe₂ nanosheets (average thickness: ~3 nm, Supplementary Fig. 5) with a lot of exposed coordinatively unsaturated Pt atoms. Only in this electrochemical exfoliation stage, the exposed surface area can be significantly enhanced. Such a result has been reported in other materials, such as NiPS₃⁷ and Sb⁸. The following heat treatment will not further exfoliate the PtTe₂ sheets so that the heated PtTe₂ nanosheets will remain the similar surface area to the original PtTe₂ sheets. In addition, in the experimental part for HER, we used the same catalyst loading amount (0.336 mg cm⁻²) for all PtTe₂ based materials.

Indeed, the total electrode activity is not only determined by the quantity of active sites, but also by the intrinsic activity of active sites⁹³. For HER, TOF and overpotential based on

electrochemically active surface area (ECSA) can be employed to reflect the intrinsic activity of the active sites in catalysts^{60, 62, 67, 70, 82, 91, 94-96}. Our results have demonstrated that the TOF (at -0.2 V vs. RHE) of PtTe₂-600 NSs is 8.21 s⁻¹, much larger than that of PtTe₂ NSs (1.84 s⁻¹), PtTe₂-200 NSs (2.59 s⁻¹), PtTe₂-400 NSs (5.59 s⁻¹). To further demonstrate the enhanced intrinsic activity of coordinatively unsaturated Pt atoms in PtTe₂, current density in LSV curve is normalized to ECSA. As shown in Fig. R2, PtTe₂-600 NSs still reveals substantially smaller HER overpotential than that of PtTe₂ NSs, PtTe₂-200 NSs, PtTe₂-400 NSs at the same current density under the same measurement conditions. From both TOF and overpotential based on ECSA, it can be concluded that the intrinsic activity of coordinatively unsaturated Pt atoms in PtTe₂ is significantly enhanced after migration of the single atomic Te vacancies to form ordered Te vacancy clusters.

In our work, to further experimentally verify the gradually improved intrinsic properties of Pt sites in PtTe₂, CV measurements of PtTe₂ NS, PtTe₂-200 NS, PtTe₂-600 NS and Pt/C have been performed in Ar purged 1.0 M KOH. As shown in Supplementary Fig. 26, the underpotentially deposited hydrogen (H_{upd}) peak of PtTe₂-600 NSs at ~-0.21 V vs. RHE is negatively shifted relative to that of PtTe₂ NSs (~-0.26 V vs. RHE) and PtTe₂-200 NSs (~-0.25 V vs. RHE), suggesting weaker adsorption of hydrogen on PtTe₂-600 NSs^{91, 97}. Additionally, the H_{upd} peak of PtTe₂-600 NSs is even more negative than that of the Pt/C catalyst. This experimental result indicates that during heat treatment, the Te-SAVs in atomically thin PtTe₂ NSs migrate to form ordered Te-SAV clusters, which effectively reduces the adsorption strength of hydrogen on the Pt sites and thus drastically enhances the hydrogen evolution performance. More importantly, we also experimentally performed scanning tunneling spectroscopy (STS, dI/dV versus V) to probe the local DOS over different vacancy structures. The results show that near-Fermi local DOS of defect-free site, Te-SAV and trigonal Te-SAV in PtTe₂ (marked to red, orange, and blue symbols, respectively, in Fig. 2d) gradually decreases (Supplementary Fig. 28c). The result is consistent with the established theory calculation, which indicates a decreased DOS of Pt sites after the clustering of regular Te-SAVs (Fig. 4d and Supplementary Fig. 27).

From the above results and discussion, it can be concluded that 18 at.% of Te defect sites and 8.7 wt.% of coordinatively unsaturated Pt atoms in PtTe₂ nanosheets are obviously enough

to influence the materials' overall catalytic activity. More importantly, experimental results also show that the intrinsic activity of coordinatively unsaturated Pt atoms in PtTe₂ is significantly enhanced after the migration of the single atomic Te vacancies to form ordered Te vacancy clusters. Thus, the theory we have built up to explain the enhanced activity is reasonable.

Fig. R2 LSV curves of PtTe₂ NSs, PtTe₂-200 NSs, PtTe₂-400 NSs, and PtTe₂-600 NSs with current normalized to ECSA.

2. The LSV curves of the samples in 1.0 M KOH (Fig. 3) clearly indicates the enhancement of HER performance with respect to temperature. It is a clear indication of enhanced crystallinity and conductivity, which enhances the activity. The improved activity is not related to defect sites; instead, it is related to crystallinity and conductivity (As the crystallinity is enhanced after annealing at 600 °C), as Pt is a well-known the most active HER catalyst.

Response: We thank the reviewer for the comments, but it is too arbitrary for the reviewer to draw such a conclusion. The LSV curves of the samples recorded in 1.0 M KOH (Fig. 3) indicate enhancement of HER performance with respect to temperature. However, this result is completely not an indication of enhanced crystallinity and conductivity. Indeed, our bulk PtTe₂ crystals are synthesized at 1000 °C for 48 h and then 1150 °C for another 1 h in furnace (details in experimental part). The obtained crystals already have good crystallinity as demonstrated by the XRD result as shown in Fig. 1c. Electrochemical exfoliation method also has been proved as an efficient strategy to produce atomically thin 2D materials with high

quality and crystallinity⁹⁸⁻¹⁰². Thus, after exfoliating the bulk PtTe₂ crystals, the obtained atomically thin PtTe₂ NSs still have good quality and crystallinity, which was already demonstrated by the high-resolution HAADF-STEM image in Fig. 1f. The following heat treatment which causes migration of the random Te-SAVs to form ordered Te-SAV clusters is below 600 °C, much lower than the crystal growth temperature (1000-1150 °C).

To further exclude the possibility that the improved activity is related to crystallinity, we conducted the following experiments. Firstly, 4 mg of the exfoliated PtTe₂ NSs were dispersed in 1 ml of ethanol with 8 μl of Nafion 117 solution and sonicated for ~2 minutes to obtain a homogeneous catalyst ink. Then, 25 μl of the catalyst ink was dropped onto two carbon papers (Toray Paper 060, 1cm × 1cm) with the same area (1cm × 0.5cm) and loading amount (0.198 mg cm⁻²). After completely drying, one carbon paper with exfoliated PtTe₂ NSs was heated under vacuum (1×10^{-9} mbar) in the STM equipment at only 120 °C for 2 h. The randomly dispersed Te-SAVs in PtTe₂ can also partially migrate and form ordered Te-SAV clusters after the low-temperature treatment in this condition (1×10^{-9} mbar, 120 °C for 2 h), as demonstrated in Fig. R3a. At last, we compared the HER performance of the two materials. Here, the treated sample is denoted as PtTe₂ NSs-120 °C in vacuum, and the original sample is denoted as PtTe₂ NSs. Obviously, these two samples have nearly similar crystallinity. As shown in Fig. R3b, the sample PtTe₂ NSs-120 °C in vacuum obviously exhibits the significantly higher HER activity as compared to PtTe₂ NSs. From the above results, it can be concluded that the improved activity should be related to the ordered clustering of single atomic Te vacancies, rather than the crystallinity.

For conductivity, it should be noted that when the thickness increases from monolayer to bilayer, PtTe₂ evolves from semiconductive (bandgap: ~0.84 eV) to metallic. The electronic structure of trilayer PtTe₂ has already resembled to its bulk material (Supplementary Figs. 19 and 20). Since metallic nature is of pivotal importance for hydrogen adsorption, we have performed the atomic force microscopy measurements, which show that the PtTe₂ NSs have thicknesses about 0.6–6 nm (average thickness: ~3 nm) (Supplementary Fig. 5). Most of the exfoliated PtTe₂ NSs have more than 3-layer structures as the theoretical thickness of the monolayer PtTe₂ is about 0.67 nm, suggesting that PtTe₂ NSs have good conductivity. Thus, the conductivity should not be the decisive factor for the significantly imported HER

performance.

As we mentioned in our reply to question 1, the intrinsic activity of coordinatively unsaturated Pt in PtTe₂ is significantly improved after the clustering of single atomic Te vacancies, which has been demonstrated by the TOF and overpotential based on electrochemically active surface area (ECSA)^{60, 62, 67, 70, 82, 91, 94-96}. The TOF (at -0.2 V vs. RHE) of PtTe₂-600 NSs is 8.21 s⁻¹, much larger than that of PtTe₂ NSs (1.84 s⁻¹), PtTe₂-200 NSs (2.59 s⁻¹), PtTe₂-400 NSs (5.59 s⁻¹). What's more, PtTe₂-600 NSs reveals substantially smaller HER overpotential than that of PtTe₂ NSs, PtTe₂-200 NSs, PtTe₂-400 NSs at the same current density under the same measurement conditions (shown in Fig. R2). The underpotentially deposited hydrogen (H_{upd}) peak of PtTe₂-600 NSs at ~0.21 V vs. RHE as shown in Supplementary Fig. 26 is negatively shifted relative to that of PtTe₂ NSs (~0.26 V vs. RHE) and PtTe₂-200 NSs (~0.25 V vs. RHE), suggesting weaker adsorption of hydrogen on PtTe₂-600 NSs^{91, 97}. Additionally, the H_{upd} peak of PtTe₂-600 NSs is even more negative than that of the Pt/C catalyst. More importantly, we experimentally performed scanning tunneling spectroscopy (STS, dI/dV versus V) to probe the local DOS over different vacancy structures. The results show that near-Fermi local DOS of defect-free site, Te-SAV and trigonal Te-SAV in PtTe₂ (marked to red, orange, and blue symbols, respectively, in Fig. 2d) gradually decreases (Supplementary Fig. 28c). The result is consistent with the established theory calculation, which indicates a decreased DOS of Pt sites after the clustering of regular Te-SAVs (Fig. 4d and Supplementary Fig. 27).

From these mutually corroborating data, it can be concluded that during heat treatment, the Te-SAVs in atomically thin PtTe₂ NSs migrate to form ordered Te-SAV clusters, which effectively reduces the adsorption strength of hydrogen on the Pt sites and thus drastically enhances the hydrogen evolution performance. The improved activity is strongly related to the significantly improved intrinsic activity of coordinatively unsaturated Pt in PtTe₂, and rather than crystallinity and conductivity (as they are already good enough).

Fig. R3 **a** STM image of PtTe₂ showing that randomly dispersed Te-SAVs in PtTe₂ can partially migrate and form ordered Te-SAV clusters after treatment in 1×10^{-9} mbar, 120 °C for 2 h. **b** LSV curves of PtTe₂ NSs, and PtTe₂ NSs-120 °C in vacuum, recorded in 1.0 M KOH, at a scan rate of 5 mV s⁻¹, with 80 % iR compensation. The working electrode is carbon paper, and the catalyst loading amount is 0.198 mg cm⁻².

3. *The straightforward comparison between Pt/C (20%) with PtTe₂ will not be fair, as the amount of Pt in PtTe₂ is much higher than the Pt/C. It means the per unit mass activity of Pt/C is much better than the PtTe₂.*

Response: We thank the reviewer for raising the question. Actually, when comparing the activity of Pt/C (20 wt.%) with PtTe₂, we kept the amount of Pt in PtTe₂ on glassy carbon electrode (GCE) the same as that in Pt/C, which means that the per unit mass activity of Pt/C is the same as that in PtTe₂. As already stated in the experimental part, before loading catalyst, GCE was successively polished with 1.0, 0.3, and 0.05 mm Al₂O₃ slurry to obtain an ultraclean surface. Afterwards, 4 mg of catalysts were dispersed in 1 ml of ethanol with 8 μl of Nafion 117 solution and sonicated for ~2 minutes to obtain a homogeneous catalyst ink. For commercial Pt/C (20 wt.%), 40 μl of the catalyst ink was dropped onto the GCE and allowed to dry at room temperature. Thus, the total mass loading of Pt is 0.162 mg cm⁻². For PtTe₂ (n_{Pt}:n_{Te}=0.61), 16.6 μl of the catalyst ink was dropped onto the GCE and dried at room temperature. The average loading of the catalyst is 0.336 mg cm⁻², and the Pt content is ~0.162 mg cm⁻²,

which is the same as that in Pt/C (20 wt.%) catalyst. The LSV curves based on mass activity of Pt in PtTe₂-600 NSs and Pt/C are shown in Fig. R4a, we can see that the mass activity of Pt in PtTe₂-600 NSs is obviously higher than that in Pt/C at the same potential. At -0.2 V vs. RHE, the mass activity of Pt in PtTe₂-600 NSs is calculated to be 1.55 A mg_{Pt}⁻¹, while that is only 1.13 A mg_{Pt}⁻¹ for Pt in Pt/C.

Fig. R4 a Comparison of LSV curves based on mass activity of Pt in PtTe₂-600 NSs and Pt/C.

b Mass activity comparison between PtTe₂-600 NSs and Pt/C catalysts at -0.2 V vs. RHE.

4. Moreover, it is not the best-reported values compared to literature available catalysts. There are many Ru and Ir based catalysts, which are better than this catalyst. The authors have only shown the catalysts which are less active their catalyst.

Response: We thank the reviewer for the comments. We agree that our PtTe₂ catalyst's HER performance is not the best in literature, and some Ru and Ir based catalysts exhibit better HER performance than our catalyst. Based on the reviewer's suggestion, we have added more Ru and Ir based catalysts with super HER activity in the Supplementary Table 2. However, as shown in Fig. 3e and Supplementary Table S2, PtTe₂ catalyst still outperforms many of the reported highly active HER catalysts, especially Pt-based catalysts. It is commonly known that many factors can affect the activity measurement of HER, such as catalyst loading amount⁵⁷⁻⁶², electrode preparation procedure, iR correction⁶³⁻⁶⁶, reference electrode calibration^{60, 61, 67-70}, temperature⁷¹⁻⁷⁴, scan speed⁷⁵, and so on. However, for example, many published papers regarding catalysts with high HER activities even do not contain the information about

reference electrode calibration, as marked in the Supplementary Table S2. In addition, some published works with very high HER activities even selected Pt wire as the counter electrode⁷⁶, which is obviously a mistake because it has been reported that the Pt electrode can be dissolved from the oxidation reaction (Pt^0 to Pt^{2+}) and the dissolved Pt^{2+} can transfer to the working electrode surface and is deposited on the electrode materials⁷⁷⁻⁷⁹. Apart from the activity, catalytic stability is equally important. Accelerated cyclic voltammetry (CV) test elucidates the robustness of PtTe_2 -600 NSs in HER catalysis with virtually unchanged polarization curves even after 20,000 CV cycles (Fig. 3h). Furthermore, long-term chronopotentiometry measurement shows a slight change of overpotential after 24 h at a constant current density of 200 mA cm^{-2} for PtTe_2 -600 NSs, much better than that of Pt/C catalyst (Fig. 3i).

As stated before, the novelty of our work is that we designed and prepared atomically thin 2D PtTe_2 nanosheets with exposed and stabilized undercoordinated Pt sites as an atomically controllable Pt-based model catalyst to understand the correlation between electronic structure, adsorption energy, and catalytic property of atomic Pt sites. Also, we found that thermal treatment could drive the migration of the well-dispersed single atomic Te vacancies in PtTe_2 to form thermodynamically stabilized, ordered trigonal well-dispersed single atomic Te vacancy clusters. This finding provides a new strategy to engineer geometrically well-defined active sites via clustering of atomic defects.

Reviewer #3 (Remarks to the Author):

My previous comments have been fully addressed. I strongly support the publication of this work.

Response: We thank the reviewer for the recommendation to publish our work.

References

1. Vancsó P, *et al.* The intrinsic defect structure of exfoliated MoS₂ single layers revealed by scanning tunneling microscopy. *Sci. Rep.* **6**, 29726 (2016).
2. Ugeda MM, *et al.* Giant bandgap renormalization and excitonic effects in a monolayer transition metal dichalcogenide semiconductor. *Nat. Mater.* **13**, 1091-1095 (2014).
3. Zhu H, *et al.* Defects and surface structural stability of MoTe₂ under vacuum annealing. *ACS Nano* **11**, 11005-11014 (2017).
4. Zhang S, *et al.* Defect structure of localized excitons in a WSe₂ monolayer. *Phys. Rev. Lett.* **119**, 046101 (2017).
5. Zheng H, *et al.* Visualization of point defects in ultrathin layered 1T-PtSe₂. *2D Mater.* **6**, 041005 (2019).
6. Gao J, *et al.* Structure, stability, and kinetics of vacancy defects in monolayer PtSe₂: A first-principles study. *ACS Omega* **2**, 8640-8648 (2017).
7. Li X, *et al.* High-yield electrochemical production of large-sized and thinly layered NiPS₃ flakes for overall water splitting. *Small* **15**, 1902427 (2019).
8. Li FW, *et al.* Unlocking the electrocatalytic activity of antimony for CO₂ reduction by two-dimensional engineering of the bulk material. *Angew. Chem. Int. Edit.* **56**, 14718-14722 (2017).
9. Xie C, *et al.* Insight into the design of defect electrocatalysts: From electronic structure to adsorption energy. *Mater. Today* **31**, 47-68 (2019).
10. Zhang T, *et al.* Engineering oxygen vacancy on NiO nanorod arrays for alkaline hydrogen evolution. *Nano Energy* **43**, 103-109 (2018).
11. Liu Y, Xiao C, Li Z, Xie Y. Vacancy engineering for tuning electron and phonon structures of two-dimensional materials. *Adv. Energy Mater.* **6**, 1600436 (2016).
12. Zhang J-J, *et al.* Oxygen vacancy engineering of Co₃O₄ nanocrystals through coupling with

- metal support for water oxidation. *ChemSusChem* **10**, 2875-2879 (2017).
13. Jiao X, *et al.* Defect-mediated electron-hole separation in one-unit-cell ZnIn₂S₄ layers for boosted solar-driven CO₂ reduction. *J. Am. Chem. Soc.* **139**, 7586-7594 (2017).
 14. Wang X, *et al.* Single-atom vacancy defect to trigger high-efficiency hydrogen evolution of MoS₂. *J. Am. Chem. Soc.* **142**, 4298-4308 (2020).
 15. Yan Y, *et al.* Tailoring the edge sites of 2d Pd nanostructures with different fractal dimensions for enhanced electrocatalytic performance. *Adv. Sci.* **5**, 1800430 (2018).
 16. Dou S, Tao L, Wang R, El Hankari S, Chen R, Wang S. Plasma-assisted synthesis and surface modification of electrode materials for renewable energy. *Adv. Mater.* **30**, 1705850 (2018).
 17. Xu L, *et al.* Plasma-engraved Co₃O₄ nanosheets with oxygen vacancies and high surface area for the oxygen evolution reaction. *Angewandte Chemie International Edition* **55**, 5277-5281 (2016).
 18. Dou S, Tao L, Huo J, Wang S, Dai L. Etched and doped Co₉S₈/graphene hybrid for oxygen electrocatalysis. *Energy Environ. Sci.* **9**, 1320-1326 (2016).
 19. Chen D, *et al.* Preferential cation vacancies in perovskite hydroxide for the oxygen evolution reaction. *Angew. Chem. Int. Ed.* **57**, 8691-8696 (2018).
 20. Gao S, *et al.* Atomic layer confined vacancies for atomic-level insights into carbon dioxide electroreduction. *Nat. Commun.* **8**, 14503 (2017).
 21. Gao S, *et al.* Highly efficient and exceptionally durable CO₂ photoreduction to methanol over freestanding defective single-unit-cell bismuth vanadate layers. *J. Am. Chem. Soc.* **139**, 3438-3445 (2017).
 22. Li L, *et al.* Role of sulfur vacancies and undercoordinated mo regions in MoS₂ nanosheets toward the evolution of hydrogen. *ACS Nano* **13**, 6824-6834 (2019).
 23. Yue Q, Liu C, Wan Y, Wu X, Zhang X, Du P. Defect engineering of mesoporous nickel ferrite and its application for highly enhanced water oxidation catalysis. *J. Catal.* **358**, 1-7 (2018).
 24. Xu L, *et al.* Plasma-engraved Co₃O₄ nanosheets with oxygen vacancies and high surface area for the oxygen evolution reaction. *Angew. Chem. Int. Edit.* **55**, 5277-5281 (2016).
 25. Wang Y, *et al.* Layered double hydroxide nanosheets with multiple vacancies obtained by

- dry exfoliation as highly efficient oxygen evolution electrocatalysts. *Angew. Chem. Int. Ed.* **56**, 5867-5871 (2017).
26. Xiao Z, *et al.* Filling the oxygen vacancies in Co_3O_4 with phosphorus: An ultra-efficient electrocatalyst for overall water splitting. *Energy Environ. Sci.* **10**, 2563-2569 (2017).
 27. Wang S, Iyyamperumal E, Roy A, Xue Y, Yu D, Dai L. Vertically aligned bcn nanotubes as efficient metal-free electrocatalysts for the oxygen reduction reaction: A synergetic effect by co-doping with boron and nitrogen. *Angew. Chem. Int. Ed.* **50**, 11756-11760 (2011).
 28. Gong K, Du F, Xia Z, Durstock M, Dai L. Nitrogen-doped carbon nanotube arrays with high electrocatalytic activity for oxygen reduction. *Science* **323**, 760 (2009).
 29. Dai L, Xue Y, Qu L, Choi H-J, Baek J-B. Metal-free catalysts for oxygen reduction reaction. *Chem. Rev.* **115**, 4823-4892 (2015).
 30. Qiu H, *et al.* Hopping transport through defect-induced localized states in molybdenum disulphide. *Nat. Commun.* **4**, 2642 (2013).
 31. Guo Y, Robertson J. Vacancy and doping states in monolayer and bulk black phosphorus. *Sci. Rep.* **5**, 14165 (2015).
 32. Wu Y, *et al.* Regulating the interfacial electronic coupling of Fe_2N via orbital steering for hydrogen evolution catalysis. *Adv. Mater.* **32**, 1904346 (2020).
 33. Fu Z, Yang B, Wu R. Understanding the activity of single-atom catalysis from frontier orbitals. *Phys. Rev. Lett.* **125**, 156001 (2020).
 34. Suntivich J, Gasteiger HA, Yabuuchi N, Nakanishi H, Goodenough JB, Shao-Horn Y. Design principles for oxygen-reduction activity on perovskite oxide catalysts for fuel cells and metal-air batteries. *Nat. Chem.* **3**, 546-550 (2011).
 35. Zhang L, *et al.* Atomic layer deposited Pt-Ru dual-metal dimers and identifying their active sites for hydrogen evolution reaction. *Nat. Commun.* **10**, 4936 (2019).
 36. Xin H, Vojvodic A, Voss J, Nørskov JK, Abild-Pedersen F. Effects of *d*-band shape on the surface reactivity of transition-metal alloys. *Phys. Rev. B* **89**, 115114 (2014).
 37. Ruban A, Hammer B, Stoltze P, Skriver HL, Nørskov JK. Surface electronic structure and reactivity of transition and noble metals. Communication presented at the first francqui colloquium, brussels, 19-20 february 1996.1. *J. Mol. Catal. A: Chem.* **115**, 421-429 (1997).
 38. Seh ZW, Kibsgaard J, Dickens CF, Chorkendorff I, Nørskov JK, Jaramillo TF. Combining

- theory and experiment in electrocatalysis: Insights into materials design. *Science* **355**, eaad4998 (2017).
39. Tian J, Liu Q, Asiri AM, Sun X. Self-supported nanoporous cobalt phosphide nanowire arrays: An efficient 3d hydrogen-evolving cathode over the wide range of pH 0–14. *J. Am. Chem. Soc.* **136**, 7587-7590 (2014).
 40. Tang C, *et al.* Fe-doped cop nanoarray: A monolithic multifunctional catalyst for highly efficient hydrogen generation. *Adv. Mater.* **29**, 1602441 (2017).
 41. Liu T, *et al.* Enhanced electrocatalysis for energy-efficient hydrogen production over cop catalyst with nonelectroactive Zn as a promoter. *Adv. Energy Mater.* **7**, 1700020 (2017).
 42. Pu Z, Liu Q, Tang C, Asiri AM, Sun X. Ni₂P nanoparticle films supported on a Ti plate as an efficient hydrogen evolution cathode. *Nanoscale* **6**, 11031-11034 (2014).
 43. Popczun EJ, *et al.* Nanostructured nickel phosphide as an electrocatalyst for the hydrogen evolution reaction. *J. Am. Chem. Soc.* **135**, 9267-9270 (2013).
 44. Tang C, *et al.* Energy-saving electrolytic hydrogen generation: Ni₂P nanoarray as a high-performance non-noble-metal electrocatalyst. *Angew. Chem. Int. Ed.* **56**, 842-846 (2017).
 45. Kibsgaard J, Jaramillo TF. Molybdenum phosphosulfide: An active, acid-stable, earth-abundant catalyst for the hydrogen evolution reaction. *Angew. Chem. Int. Ed.* **53**, 14433-14437 (2014).
 46. Xing Z, Liu Q, Asiri AM, Sun X. Closely interconnected network of molybdenum phosphide nanoparticles: A highly efficient electrocatalyst for generating hydrogen from water. *Adv. Mater.* **26**, 5702-5707 (2014).
 47. Li Y, *et al.* Partially exposed RuP₂ surface in hybrid structure endows its bifunctionality for hydrazine oxidation and hydrogen evolution catalysis. *Sci. Adv.* **6**, eabb4197 (2020).
 48. Tian JQ, Liu Q, Cheng NY, Asiri AM, Sun XP. Self-supported Cu₃P nanowire arrays as an integrated high-performance three-dimensional cathode for generating hydrogen from water. *Angew. Chem. Int. Edit.* **53**, 9577-9581 (2014).
 49. Jiang P, Liu Q, Liang YH, Tian JQ, Asiri AM, Sun XP. A cost-effective 3d hydrogen evolution cathode with high catalytic activity: FeP nanowire array as the active phase. *Angew. Chem. Int. Edit.* **53**, 12855-12859 (2014).
 50. Laursen AB, *et al.* Nanocrystalline Ni₅P₄: A hydrogen evolution electrocatalyst of

- exceptional efficiency in both alkaline and acidic media. *Energy Environ. Sci.* **8**, 1027-1034 (2015).
51. Liu P, Rodriguez JA. Catalysts for hydrogen evolution from the [NiFe] hydrogenase to the Ni₂P(001) surface: The importance of ensemble effect. *J. Am. Chem. Soc.* **127**, 14871-14878 (2005).
 52. Shi Y, Zhang B. Recent advances in transition metal phosphide nanomaterials: Synthesis and applications in hydrogen evolution reaction. *Chem. Soc. Rev.* **45**, 1529-1541 (2016).
 53. Yang Y, Xue X-X, Chen Q-j, Feng Y. Doping single transition metal atom into PtTe sheet for catalyzing nitrogen reduction and hydrogen evolution reactions. *J. Chem. Phys.* **151**, 144710 (2019).
 54. Liu S, *et al.* Cation vacancy-modulated PtPdRuTe five-fold twinned nanomaterial for catalyzing hydrogen evolution reaction. *Nano Energy* **61**, 346-351 (2019).
 55. Bai L, *et al.* Highly efficient utilization of precious metals for hydrogen evolution reaction with photo-assisted electro-deposited urchin-like Te nanostructure as a template. *ChemCatChem* **11**, 2283-2287 (2019).
 56. Yan M, *et al.* Lorentz-violating type-ii dirac fermions in transition metal dichalcogenide PtTe₂. *Nat. Commun.* **8**, 257 (2017).
 57. Wu L, *et al.* Stable cobalt nanoparticles and their monolayer array as an efficient electrocatalyst for oxygen evolution reaction. *J. Am. Chem. Soc.* **137**, 7071-7074 (2015).
 58. Xing M, Kong L-B, Liu M-C, Liu L-Y, Kang L, Luo Y-C. Cobalt vanadate as highly active, stable, noble metal-free oxygen evolution electrocatalyst. *J. Mater. Chem. A* **2**, 18435-18443 (2014).
 59. Lao M, *et al.* Platinum/nickel bicarbonate heterostructures towards accelerated hydrogen evolution under alkaline conditions. *Angew. Chem. Int. Ed.* **58**, 5432-5437 (2019).
 60. Mao J, *et al.* Accelerating water dissociation kinetics by isolating cobalt atoms into ruthenium lattice. *Nat. Commun.* **9**, 4958 (2018).
 61. Li Z, Fu J-Y, Feng Y, Dong C-K, Liu H, Du X-W. A silver catalyst activated by stacking faults for the hydrogen evolution reaction. *Nat. Catal.* **2**, 1107-1114 (2019).
 62. Anantharaj S, Kundu S. Do the evaluation parameters reflect intrinsic activity of electrocatalysts in electrochemical water splitting? *ACS Energy Letters* **4**, 1260-1264

- (2019).
63. Gong M, *et al.* An advanced Ni–Fe layered double hydroxide electrocatalyst for water oxidation. *J. Am. Chem. Soc.* **135**, 8452-8455 (2013).
 64. Ma TY, Dai S, Jaroniec M, Qiao SZ. Graphitic carbon nitride nanosheet-carbon nanotube three-dimensional porous composites as high-performance oxygen evolution electrocatalysts. *Angew. Chem. Int. Edit.* **53**, 7281-7285 (2014).
 65. Duan J, Chen S, Ortíz-Ledón CA, Jaroniec M, Qiao S-Z. Phosphorus vacancies that boost electrocatalytic hydrogen evolution by two orders of magnitude. *Angew. Chem. Int. Ed.* **59**, 8181-8186 (2020).
 66. Huang Y, *et al.* Atomically engineering activation sites onto metallic 1T-MoS₂ catalysts for enhanced electrochemical hydrogen evolution. *Nat. Commun.* **10**, 982 (2019).
 67. Mahmood J, *et al.* An efficient and pH-universal ruthenium-based catalyst for the hydrogen evolution reaction. *Nat. Nanotechnol.* **12**, 441-446 (2017).
 68. Liu D, *et al.* Atomically dispersed platinum supported on curved carbon supports for efficient electrocatalytic hydrogen evolution. *Nat. Energy* **4**, 512-518 (2019).
 69. Liang Y, *et al.* Co₃O₄ nanocrystals on graphene as a synergistic catalyst for oxygen reduction reaction. *Nat. Mater.* **10**, 780-786 (2011).
 70. Lu B, *et al.* Ruthenium atomically dispersed in carbon outperforms platinum toward hydrogen evolution in alkaline media. *Nat. Commun.* **10**, 631 (2019).
 71. Tang Z-q, Liao L-w, Zheng Y-l, Kang J, Chen Y-x. Temperature effect on hydrogen evolution reaction at au electrode. *Chin. J. Chem. Phys.* **25**, 469-474 (2012).
 72. Gennero de Chialvo MR, Chialvo AC. Hydrogen evolution reaction on a smooth iron electrode in alkaline solution at different temperatures. *Physical Chemistry Chemical Physics* **3**, 3180-3184 (2001).
 73. Pierozynski B, Mikolajczyk T. On the temperature dependence of hydrogen evolution reaction at nickel foam and Pd-modified nickel foam catalysts. *Electrocatalysis* **6**, 51-59 (2015).
 74. Vračar LM, Dražić DM. Anomalous temperature dependence of the hydrogen evolution reaction on iron. *J. Electroanal. Chem. Interfacial Electrochem.* **265**, 171-178 (1989).
 75. Wang Q, *et al.* Coordination engineering of iridium nanocluster bifunctional electrocatalyst

- for highly efficient and pH-universal overall water splitting. *Nat. Commun.* **11**, 4246 (2020).
76. Ye R, *et al.* High performance electrocatalytic reaction of hydrogen and oxygen on ruthenium nanoclusters. *ACS Applied Materials & Interfaces* **9**, 3785-3791 (2017).
77. Chen R, *et al.* Use of platinum as the counter electrode to study the activity of nonprecious metal catalysts for the hydrogen evolution reaction. *ACS Energy Letters* **2**, 1070-1075 (2017).
78. Wang J, Xu F, Jin H, Chen Y, Wang Y. Non-noble metal-based carbon composites in hydrogen evolution reaction: Fundamentals to applications. *Adv. Mater.* **29**, 1605838 (2017).
79. Dong G, *et al.* Insight into the electrochemical activation of carbon-based cathodes for hydrogen evolution reaction. *J. Mater. Chem. A* **3**, 13080-13086 (2015).
80. Meng C, Lin M-C, Du X-W, Zhou Y. Molybdenum disulfide modified by laser irradiation for catalyzing hydrogen evolution. *ACS Sustainable Chem. Eng.* **7**, 6999-7003 (2019).
81. Tsai C, *et al.* Electrochemical generation of sulfur vacancies in the basal plane of MoS₂ for hydrogen evolution. *Nat. Commun.* **8**, 15113 (2017).
82. Li H, *et al.* Activating and optimizing MoS₂ basal planes for hydrogen evolution through the formation of strained sulphur vacancies. *Nat. Mater.* **15**, 48-53 (2016).
83. Xia B, *et al.* Ar²⁺ beam irradiation-induced multivacancies in MoSe₂ nanosheet for enhanced electrochemical hydrogen evolution. *ACS Energy Letters* **3**, 2167-2172 (2018).
84. Tao HB, *et al.* Identification of surface reactivity descriptor for transition metal oxides in oxygen evolution reaction. *J. Am. Chem. Soc.* **138**, 9978-9985 (2016).
85. Sun Y, Gao S, Lei F, Xie Y. Atomically-thin two-dimensional sheets for understanding active sites in catalysis. *Chem. Soc. Rev.* **44**, 623-636 (2015).
86. Sun Y, Gao S, Xie Y. Atomically-thick two-dimensional crystals: Electronic structure regulation and energy device construction. *Chem. Soc. Rev.* **43**, 530-546 (2014).
87. Sun Y, *et al.* Pits confined in ultrathin cerium(IV) oxide for studying catalytic centers in carbon monoxide oxidation. *Nat. Commun.* **4**, 2899 (2013).
88. Tao L, *et al.* Creating coordinatively unsaturated metal sites in metal-organic-frameworks as efficient electrocatalysts for the oxygen evolution reaction: Insights into the active centers. *Nano Energy* **41**, 417-425 (2017).

89. Yan C, *et al.* Coordinatively unsaturated nickel–nitrogen sites towards selective and high-rate CO₂ electroreduction. *Energy Environ. Sci.* **11**, 1204-1210 (2018).
90. Fang S, *et al.* Uncovering near-free platinum single-atom dynamics during electrochemical hydrogen evolution reaction. *Nat. Commun.* **11**, 1029 (2020).
91. Jiang K, *et al.* Single platinum atoms embedded in nanoporous cobalt selenide as electrocatalyst for accelerating hydrogen evolution reaction. *Nat. Commun.* **10**, 1743 (2019).
92. Zhang J, *et al.* Single platinum atoms immobilized on an Mxene as an efficient catalyst for the hydrogen evolution reaction. *Nat. Catal.* **1**, 985-992 (2018).
93. Kou T, *et al.* Carbon doping switching on the hydrogen adsorption activity of NiO for hydrogen evolution reaction. *Nat. Commun.* **11**, 590 (2020).
94. Xue Y, *et al.* Anchoring zero valence single atoms of nickel and iron on graphdiyne for hydrogen evolution. *Nat. Commun.* **9**, 1460 (2018).
95. Kibsgaard J, Jaramillo TF, Besenbacher F. Building an appropriate active-site motif into a hydrogen-evolution catalyst with thiomolybdate [Mo₃S₁₃]²⁻ clusters. *Nat. Chem.* **6**, 248-253 (2014).
96. Zang Y, *et al.* Tuning orbital orientation endows molybdenum disulfide with exceptional alkaline hydrogen evolution capability. *Nat. Commun.* **10**, 1217 (2019).
97. Luo M, *et al.* PdMo bimetallic for oxygen reduction catalysis. *Nature* **574**, 81-85 (2019).
98. Lin Z, *et al.* Solution-processable 2d semiconductors for high-performance large-area electronics. *Nature* **562**, 254-258 (2018).
99. Li J, *et al.* Printable two-dimensional superconducting monolayers. *Nat. Mater.*, 10.1038/s41563-41020-00831-41561 (2020).
100. Yu W, *et al.* Chemically exfoliated VSe₂ monolayers with room-temperature ferromagnetism. *Adv. Mater.* **31**, 1903779 (2019).
101. Li J, *et al.* Ultrafast electrochemical expansion of black phosphorus toward high-yield synthesis of few-layer phosphorene. *Chem. Mater.* **30**, 2742-2749 (2018).
102. Li J, *et al.* Printable two-dimensional superconducting monolayers. *Nat. Mater.*, 10.1038/s41563-41020-00831-41561 (2020).

REVIEWER COMMENTS

Reviewer #1 (Remarks to the Author):

The revised paper is ok for acceptance.

Reviewer #2 (Remarks to the Author):

The authors are highly appreciated for their effort to revise the manuscript based on the reviewers' comments. Unfortunately, however, it is not clear enough to recommend for publication.

1. First, the way the authors have described GC electrode preparation of the Pt/C and PtTe₂ clearly indicating the miss interpretation of the PtTe₂'s performance. The authors have dropped 40 μl of Pt/C, which is about 0.809 mg cm^{-2} , and the mass loading of Pt is 0.162 mg cm^{-2} . Depositing this much of Pt/C on GCE is itself a big mistake. Based on the reviewer's experience, when the loading of Pt/C goes above 0.7 mg cm^{-2} , the activity sharply goes down, because of the too thick and rough film formation.

2. Second, for the per unit mass activity, the authors have equalized the mass loading of Pt (0.162 mg cm^{-2}) of Pt/C, about 16.6 μl PtTe₂ catalyst ink, and loading of the catalyst is 0.336 mg cm^{-2} , which is relatively less compared to the 0.809 mg cm^{-2} of Pt/C. The per unit mass activity is calculated from the LSV after using the reference and the sample's equal mass on the electrode. This mistake itself enough to decline the consideration of this manuscript.

3. Third, the development of a new catalyst should be based on activity, stability, and cost. The use of precious metal with such high loading is meaningless, if it is more than the commercial Pt/C.n of this work.

Point-by-point response letter

We thank the reviewers for their valuable comments (*words in italics*), response to which will undoubtedly improve our manuscript and clarify issues we may have failed to point out in the last version. We have provided our point-by-point response to each comment (in blue). The revised part in the Main Text and Supporting Information are marked with blue.

Reviewer #2 (Remarks to the Author):

The authors are highly appreciated for their effort to revise the manuscript based on the reviewers' comments. Unfortunately, however, it is not clear enough to recommend for publication.

1. First, the way the authors have described GC electrode preparation of the Pt/C and PtTe₂ clearly indicating the miss interpretation of the PtTe₂'s performance. The authors have dropped 40 μl of Pt/C, which is about 0.809 mg cm^{-2} , and the mass loading of Pt is 0.162 mg cm^{-2} . Depositing this much of Pt/C on GCE is itself a big mistake. Based on the reviewer's experience, when the loading of Pt/C goes above 0.7 mg cm^{-2} , the activity sharply goes down, because of the too thick and rough film formation.

Response: We thank the reviewer for the comment. We agree with the reviewer that catalyst loading amount on GC electrode can affect the electrochemical performance. To investigate the influence of Pt/C (20 wt.%, Alfa Aesar) catalyst loading amount on HER, we carefully loaded commercial Pt/C catalyst with different catalyst loading amount on GC electrode (geometry area: 0.196 cm^2). Firstly, GC electrode was successively polished with 1.0, 0.3, and 0.05 mm Al_2O_3 slurry to obtain an ultraclean surface. Afterwards, 4 mg of Pt/C was dispersed in 1 ml of ethanol with 8 μl of Nafion 117 solution and sonicated for ~10 minutes to obtain a homogeneous catalyst ink. Then, X (X=0, 20, 25, 30, 35, 40, 45, 50, 60) μl of the catalyst ink was dropped onto the GC electrode and allowed to dry at room temperature. Thus, the loading amount of Pt/C on GC electrode could be carefully varied as shown in Table R1.

All of the GC electrodes were then tested for linear sweep voltammetry as detailed in the experimental part, and the catalyst loading-dependent current density at a particular overpotential was studied. As shown in Supplementary Fig. 31, the obtained data with mean deviation are plotted as catalyst loading amount vs. current density at a 50 mV overpotential. It can be seen that when the loading amount of Pt/C increases from 0.4 mg cm^{-2} to 0.9 mg cm^{-2} , the current density almost increases linearly (blue dash line in Supplementary Fig. 31). At this stage, the number of active sites dominates the HER process, rather than other factors (e.g., mass transport, conductivity). At the loading amount of Pt/C of 1.0 mg cm^{-2} , the current density starts to decrease sharply because of the too thick and rough film formation. From these results,

it can be concluded that depositing 40 μl of Pt/C (0.809 mg cm^{-2}) on GC electrode is reasonable.

Table R1. Preparation of Pt/C catalyst on GC electrode with different catalyst loading amount.

X(μl)	0	20	25	30	35	40	45	50	60
Loading amount ($\text{mg}_{\text{cat.}} \text{ cm}^{-2}$)	0	0.4	0.5	0.6	0.7	0.8	0.9	1.0	1.2

Supplementary Figure 31. Plot of catalyst loading amount vs. current density at a fixed overpotential of 50 mV. Error bar represents the mean deviation.

2. Second, for the per unit mass activity, the authors have equalized the mass loading of Pt (0.162 mg cm^{-2}) of Pt/C, about $16.6 \mu\text{l}$ PtTe₂ catalyst ink, and loading of the catalyst is 0.336 mg cm^{-2} , which is relatively less compared to the 0.809 mg cm^{-2} of Pt/C. The per unit mass activity is calculated from the LSV after using the reference and the sample's equal mass on the electrode. This mistake itself enough to decline the consideration of this manuscript.

Response: We thank the reviewer for the comments. In our work, the reason why we used 40 μl of Pt/C catalyst ink (corresponding to 0.809 mg cm^{-2}) and $16.6 \mu\text{l}$ of PtTe₂ catalyst ink (corresponding to 0.336 mg cm^{-2}) is to keep the loading amount of Pt on GC electrode the same at $0.162 \text{ mg}_{\text{Pt}} \text{ cm}^{-2}$. Thus, the per unit Pt mass activity comparison between Pt/C (20%)

and PtTe₂ is fair. This kind of mass activity comparison based on the same Pt mass is commonly used in other works, such as *Science* **354**, 1410 (2016); *Science* **348**, 1230 (2015); *Nat. Commun.* **8**, 14580 (2017); *Nat. Commun.* **7**, 11850 (2016); *Angew. Chem.* **128**, 13051 - 13055 (2016).

In our previous response letter, the per unit mass activity of **Pt (rather than the catalyst)** was calculated from the LSV by using the **equal Pt mass** on the electrode, **rather than using equal sample mass**, as shown in Supplementary Fig. 17. This calculation method for the per unit mass activity of **Pt** is also commonly adopted in many references, such as *Science* **354**, 1410 (2016); *Science* **348**, 1230 (2015); *Nat. Commun.* **7**, 11850 (2016); *Nat. Energy* **4**, 512-518 (2019); *Nat. Commun.* **10**, 1743 (2019); *Nat. Catal.* **1**, 985-992 (2018).

Supplementary Figure 17 a Comparison of LSV curves based on mass activity of Pt in PtTe₂-600 NSs and Pt/C. **b** Mass activity of Pt comparison between PtTe₂-600 NSs and Pt/C catalysts at -0.2 V vs. RHE. The mass activity of Pt in PtTe₂-600 NSs is obviously higher than that in Pt/C at the same potential. At -0.2 V vs. RHE, the mass activity of Pt in PtTe₂-600 NSs is calculated to be $1.55 \text{ A mg}^{-1}_{\text{Pt}}$, while that is only $1.13 \text{ A mg}^{-1}_{\text{Pt}}$ for Pt in Pt/C.

3. Third, the development of a new catalyst should be based on activity, stability, and cost. The use of precious metal with such high loading is meaningless, if it is more than the commercial Pt/C.

Response: We appreciate the reviewer's suggestion that activity, stability, and cost should be

considered when developing a new catalyst. The results in Fig. 3e, and Supplementary Table S2 show that PtTe₂ catalyst exhibits good activity. Furthermore, the chronopotentiometry measurement (Fig. 3i) also shows a superior stability of PtTe₂ catalyst than that of Pt/C (20 wt.%) catalyst. Although the mass fraction of Pt in PtTe₂ is higher than that in Pt/C (20 wt.%), the loading amount of Pt from PtTe₂ on the GC electrode is the same as that from Pt/C during the HER test. Moreover, as shown in Supplementary Fig. 17, the per unit mass activity of Pt in PtTe₂ is higher than that in Pt/C (20 wt.%), which means that PtTe₂ exhibits higher intrinsic catalytic activity than Pt/C.

As well known, Pt-based catalysts play versatile roles in energy-related electrocatalysis, such as alcohol oxidation reaction, hydrogen evolution/oxidation reaction, oxygen reduction reaction. Apart from activity, stability, and cost, preparation of atomically controllable Pt-based model catalysts with exposed and stabilized undercoordinated Pt sites is still of grand challenge, yet urgently required to understand the correlation between electronic structure, adsorption energy, and catalytic properties of atomic Pt sites. In this work, we designed and prepared atomically thin 2D PtTe₂ nanosheets with exposed and stabilized undercoordinated Pt sites as an atomically controllable Pt-based model catalyst to understand the correlation between electronic structure, adsorption energy, and catalytic property of atomic Pt sites. Also, we found that thermal treatment could drive migration of the well-dispersed single atomic Te vacancies in PtTe₂ to form thermodynamically stabilized, ordered trigonal well-dispersed single atomic Te vacancy clusters. This finding provides a new strategy to engineer geometrically well-defined active sites via the clustering of atomic defects, which have already been commended by other reviewers. We believe that this work will be of great interest to the broad readership, especially in view of the first report on engineering geometrically well-defined active sites via clustering of atomic defects, which allows for the unprecedented understanding of the correlations between electronic structure of catalytic center and catalytic performance.

REVIEWER COMMENTS

Reviewer #2 (Remarks to the Author):

This manuscript reports the preparation of atomically thin two-dimensional PtTe₂ nanosheets having well dispersed single atomic Te vacancies (Te-SAVs) and Pt sites as a catalyst. The Te-SAVs were created by electrochemical exfoliating bulk PtTe₂ crystals to expose the undercoordinated and stabilized Pt sites. The PtTe₂ nanosheets with ordered clusters of Te-SAVs exhibit hydrogen evolution reaction (HER) activity at low overpotential and Tafel slope.

PtTe based catalyst has already been reported in Nano Energy for HER (Nano Energy 2019, 61, 346–351347), and moreover, the catalytic activity of the given catalyst is not the best among the reported noble metal-based catalysts. In brief, this study lacks the desired originality both in catalyst design and activity, so the manuscript is not suitable for publication in Nature Communications.

1. It seems the occurrence of vacancies are significantly less in the sheet. It means the number of defects sites are relatively sparse to influence the materials' overall activity. This situation overshadows the theory the authors have built up to explain the enhanced activity. The activity could be just because of the enhanced exposed surface area.

2. The LSV curves of the samples in 1.0 M KOH (Fig. 3) clearly indicates the enhancement of HER performance with respect to temperature. It is a clear indication of enhanced crystallinity and conductivity, which enhances the activity. The improved activity is not related to defect sites; instead, it is related to crystallinity and conductivity (As the crystallinity is enhanced after annealing at 600 °C), as Pt is a well known the most active HER catalyst.

3. The straightforward comparison between Pt/C (20%) with PtTe₂ will not be fair, as the amount of Pt in PtTe₂ is much higher than the Pt/C. It means the per unit mass activity of Pt/C is much better than the PtTe₂.

4. Moreover, it is not the best-reported values compared to literature available catalysts. There are many Ru and Ir based catalysts, which are better than this catalyst. The authors have only shown the catalysts which are less active their catalyst.